# FragFormer: A Fragment-based Representation Learning Framework for Molecular Property Prediction

**Jiaxi Wang**                                                                                                        *wjx20@mails.tsinghua.edu.cn*
*Department of Electronic Engineering, Tsinghua University*

**Yaosen Min**                                                                                                        *yaosenmin@microsoft.com*
*Microsoft Research Asia*

**Miao Li***                                                                                                              *miao-li@tsinghua.edu.cn*
*Department of Electronic Engineering, Tsinghua University*

**Ji Wu***                                                                                                                *wuji__ee@tsinghua.edu.cn*
*Department of Electronic Engineering, Tsinghua University*
*College of AI, Tsinghua University*
*Beijing National Research Center for Information Science and Technology*

**Reviewed on OpenReview:** *https://openreview.net/forum?id=9aiuB3kIjd*

## Abstract

Molecular representation learning is central to molecular property prediction, which is a vital component in drug discovery. Existing methods, which mainly focus on the atom-level molecular graphs, often find it challenging to directly model the relation between fragment (substructure) and function of molecules, largely due to insufficient fragment priors. In this work, we propose a molecular self-supervised learning framework **FragFormer**, which aims to learn the representation of fragments and their contextual relationships. Given the prior that an atom can be part of multiple functional groups, we develop $k$-**D**egree **Ove**rlapping fragmentation (**DOVE**), which generates overlapping fragment graph by employing the iterative line graph. Besides, DOVE can preserve the connection information during the fragmentation phase compared to non-overlapping fragmentation. In the pre-training stage, we design a *nested masked fragment prediction* objective, to capture the hierarchical nature of fragments, namely that larger fragments can encompass multiple smaller ones. Based on FragFormer, we introduce a simple yet efficient *fragment-level* interpretation method **FragCAM** for the molecular property prediction results with greater accuracy. Moreover, thanks to the fragment modeling, our model is more capable of processing large molecule, such as peptides, and capturing the long-range interactions inside molecules. Our approach achieves state-of-the-art (SOTA) performance on eight out of eleven molecular property prediction datasets on PharmaBench. On long-range biological benchmark with peptide data, FragFormer can beat strong baselines by a clear margin, which shows the model's potential to generalize to larger molecules. Finally, we demonstrate that our model can effectively identify decisive fragments for prediction results on a real-world dataset[1].

## 1 Introduction

Finding molecules with the desired properties is one of the biggest challenges in drug discovery (Dickson & Gagnon, 2004). Using traditional wet-lab experiments to assess molecular properties is time-consuming, labor-intensive, and costly (Mullard, 2014; Simoens & Huys, 2021; Wouters et al., 2020). Machine learning models

---

*Corresponding authors.

[1]Our code is available at `https://github.com/wjxts/FragFormer`

have been developed to predict molecular properties, which can significantly reduce the cost and time (Walters & Barzilay, 2020; Wieder et al., 2020). However, the amount of data for labeled molecular properties is often limited (Guo et al., 2021) compared with the large space of pharmacologically-relevant molecules (Virshup et al., 2013), making the generalization performance of machine learning models unsatisfactory. Fortunately, a wealth of unlabeled molecular data is available (Gaulton et al., 2017; Kim et al., 2022; Sterling & Irwin, 2015). Therefore, many studies investigate self-supervised learning (SSL) methods for molecular representation learning and achieve better results than supervised learning methods (Rong et al., 2020; You et al., 2020; Zhou et al., 2023; Li et al., 2023). These SSL methods primarily focus on the atom-level molecular graphs. In pharmaceutical chemistry, fragments serve as functional groups within molecules, fundamentally influencing their properties (Guvench, 2016). Methods based on atom-level molecular graphs often suffer from over-smoothing issue (Rusch et al., 2023) and lack fragment priors (Jiang et al., 2023), making it difficult to capture the relation between fragments and the properties of molecules. Several works develop fragment-biased GNNs (Zhu et al., 2023; Bouritsas et al., 2023; Wollschläger et al., 2024) from a theoretical perspective. Nevertheless, these models primarily aim at passing certain Weisfeiler & Leman (WL) test (Zhang et al., 2023) and are not designed for molecular property prediction. There are also studies (Zhang et al., 2021b; Luong & Singh, 2023; Jiang et al., 2023) that explore the use of fragment-level molecular graphs to develop deep learning models for predicting molecular properties and achieve encouraging results. However, their molecular fragmentation methods lack overlap between fragments, overlooking the fact that a single atom can belong to different functional groups (Merlot et al., 2003). Moreover, non-overlapping fragmentation hinders their fragment graphs from distinguishing the connections between adjacent fragments. These limitations impede the model's expressivity and generalization ability. Beyond prediction accuracy, it is also important to understand the model's decision-making process. Existing interpretation methods for molecular property prediction are either atom-level (Yuan et al., 2023) or slow to achieve fragment-level interpretation (Yuan et al., 2021). In this work, we propose FragFormer, a fragment-based molecular representation learning framework to tackle the above challenges. Our contributions can be summarized as follows:

- Given the prior that an atom can be part of multiple functional groups, we propose a novel $k$-**D**egree **Ove**rlapping fragmentation (DOVE) method. DOVE can generate overlapping fragment graph by utilizing the iterative line graph transformation of the molecular graph and existing non-overlapping fragmentation methods. Besides, DOVE can retain the connection information in the fragmentation step compared to non-overlapping fragmentation.

- Taking the hierarchical nature of fragment into consideration, we design a nested masked fragment prediction self-supervised objective to model this prior.

- Our FragFormer consistently surpasses previous fragment-based models and achieves SOTA performance on eight out of eleven molecular property prediction datasets on PharmaBench.

- Based on the fragment modeling, our model is more capable of processing large molecules, outperforms strong baselines on long-range biological benchmark with peptide data.

- Building on fragment modeling, we introduce a simple fragment-level interpretation technique, called FragCAM, for the prediction outcomes. Our method achieves greater accuracy and faster speed on a real-world molecular property prediction dataset with labeled decisive fragments.

## 2 Related Works

### 2.1 Molecular Representation Learning

Obtaining good molecular representation is a crucial step for accurately predicting molecular properties (Tkatchenko, 2020). Conventional quantitative structure–activity relationship (QSAR) methods (Hansch et al., 1962; Tropsha, 2010) formulate a variety of handcrafted molecular descriptors to serve as fixed features for molecules. These features can be used as input for various machine learning models, such as SVM (Cortes & Vapnik, 1995), XGBoost (Chen & Guestrin, 2016), etc. Such fixed features can reflect simple substructures and physicochemical properties of molecules, but may not be able to capture the necessary and sophisticated

structure feature of molecules for a specific task. To adaptively learn the molecular features, supervised deep learning methods are proposed to learn the molecular representation from the molecular graph or SMILES (Weininger, 1988) string directly with labeled data (Chen et al., 2018). These methods can effectively capture the relevant features of molecular properties and outperform traditional methods when provided with sufficient labeled data (Masters et al., 2023). However, the performance of these methods is constrained by the amount of labeled data, which is usually limited in practice (Guo et al., 2021). Owing to the success of self-supervised learning (SSL) in the field of computer vision (CV) (He et al., 2022; Chen et al., 2020b), natural language processing (NLP) (Devlin et al., 2019) and audio analysis (Baevski et al., 2020; Hsu et al., 2021), SSL methods have been proposed for learning molecular representation by harnessing the abundant unlabeled molecular data. These methods primarily fall into two categories: those based on masked autoencoders, and those that employ contrastive learning. MolBERT (Fabian et al., 2020) and MolGPT (Bagal et al., 2022) use the string representation of molecules and predict the masked tokens in the SMILES string in the pre-training stage. GraphLoG (Xu et al., 2021), GROVER (Rong et al., 2020) and GEM (Fang et al., 2021) treat the molecule as 2D graph, and predict the masked nodes in the graph with graph neural networks. KPGT (Li et al., 2023) further improves the performance by modeling the global information of the molecular graph with a transformer style architecture. Contrastive learning-based methods depend on constructing different views of the same molecule. GraphCL (You et al., 2020; 2021) achieves this by randomly dropping, perturbing, and masking parts of the molecular graph. MoleculeSTM (Liu et al., 2023) utilizes text descriptions of the molecule as an alternative view, while GraphFP (Luong & Singh, 2023) and Holi-Mol (Kim et al., 2024) employ the fragment graph of the molecule for this purpose. Although these methods have advanced the performance of molecular property prediction, they mainly focus on the atom-level molecular graphs, and often struggle to directly model the relation between fragments and molecular properties due to the lack of fragment prior, restricting their interpretability and generalization. In this work, we propose a novel overlapping fragmentation method, along with a nested masked fragment prediction pre-training task, to effectively learn the representation of fragments and their contextual relationships.

## 2.2 Fragment-based Molecular Learning

Fragment-based drug discovery (FBDD) is an emerging method in drug discovery (Murray & Rees, 2009). Unlike high-throughput screening (Macarron et al., 2011), FBDD focuses on the interactions between small fragments and target proteins, and extends the fragments to larger molecules with higher binding affinity. FBDD can explore the chemical space more efficiently and reduce the cost of drug discovery with higher success rate (Murray & Rees, 2009). Motivated by the success of FBDD, researchers have proposed fragment-based methods in molecular learning tasks, mostly on molecule generation. Compared with atom-based molecule generation, fragment-based methods can generate molecules more efficiently and with more validity. Most fragment-based methods in molecule generation employ a variational autoencoder (VAE) framework (Voloboev, 2024; Kingma & Welling, 2014; Jin et al., 2018; 2020; Kong et al., 2022; Yu & Yu, 2024). JT–VAE decomposes the molecule into a tree structure (Rarey & Dixon, 1998) and generates the molecule by a tree expansion. HierVAE constructs fragment with multiple properties and subsequently expand this fragment. PS–VAE employs a data-driven method to build fragment vocabulary and generate molecules one fragment at a time, which emulates the subword vocabulary construction and tokenization process in NLP (Sennrich et al., 2016). Recently, there are some studies that explore fragment-based methods for molecular property prediction (Zhang et al., 2021a;b; Luong & Singh, 2023; Jiang et al., 2023; Kim et al., 2024). Fragment-based methods take advantage of the prior that fragments are the functional groups within molecules (Guvench, 2016), which are basic building blocks in pharmaceutical chemistry. MGSSL (Zhang et al., 2021a) utilizes predefined motifs as supervision in its self-supervised learning task. FragGAT (Zhang et al., 2021b) segments the molecular graph in different ways, processes each segment with an Attentive FP network (Xiong et al., 2020), and aggregates the segment embedding to predict the molecular properties. GraphFP (Luong & Singh, 2023) employs the fragment graph of the molecule as an alternative view in contrastive learning to enhance atom-level molecular modeling. Holi-Mol(Kim et al., 2024) further utilizes multiple views of fragmentation to enrich the representation. PharmHGT (Jiang et al., 2023) integrates both fragment and atom graphs to model the molecular properties. Although existing fragment-based methods have demonstrated encouraging results in molecular property prediction and heightened the interpretability of the model, they have yet to surpass atom-level molecular graph based methods (Chen et al., 2024). We postulate that the reason is

their fragmentation methods primarily focus on the non-overlapping fragmentation, which cannot model the fact that a single atom can contribute to different functional groups (Merlot et al., 2003). Besides, the non-overlapping fragmentation results in connection information loss between fragments. In the fragment graph with non-overlapping fragmentation, an edge cannot specify how two fragments are connected, as any two atoms within each fragment may serve as the connection points. In this work, we propose a novel overlapping fragmentation method with tunable overlapping degree and learnable fragment vocabulary. Our method allows a single atom to participate in multiple fragments and the overlapped atoms maintain the connection information between fragments.

### 2.3 Interpretation Methods for Molecular Property Prediction

Identifying the critical fragments for prediction results is essential for applying machine learning models in real-world scenarios for molecular property prediction (Proietti et al., 2024). Existing interpretation methods can be divided into two categories: atom-level and fragment-level. Atom-level interpretation methods aim to identify the important atoms for prediction results. Gradient-based attribution methods from CV, such as class activation maps (CAM) (Zhou et al., 2016), GradCAM (Selvaraju et al., 2020), GradInput (Shrikumar et al., 2017) and Integrated Gradients (IG) (Sundararajan et al., 2017), can be directly applied to score the importance of atoms in molecular property prediction tasks. Rao et al. (2021) builds a benchmark DrugXAI for interpreting molecular property prediction models and shows these methods can achieve good performance on synthetic datasets, but poor performance on real-world datasets. They also find that GradInput and IG perform better than other gradient-based methods. Many interpretation methods have been specifically developed for graph neural networks (Ying et al., 2019; Luo et al., 2020; Yuan et al., 2021). GNNExplainer (Ying et al., 2019) learns masks for node features and edges by maximizing the mutual information between the masked subgraph and the prediction. PGExplainer (Luo et al., 2020) utilizes a deep network to model the explanation generation process. While atom-level interpretation methods can highlight key atoms relevant to the predictions, they do not necessarily guarantee fragment-level explanations. SubgraphX (Yuan et al., 2021) aims to identify the most important connected subgraph for prediction results using Monte Carlo Tree Search (MCTS), offering better fragment-level interpretation compared to GNNExplainer and PGExplainer. However, it involves greater computational overhead. Existing methods either fail to guarantee fragment explanations or are slow in producing them. In this work, we propose a simple but efficient fragment-level interpretation method based on fragment modeling. Our method achieves better performance with faster speed on a real-world mutagenicity dataset.

## 3 Methods

In this section, we introduce our self-supervised learning framework, FragFormer, designed for molecular property prediction based on fragment modeling. The overall pipeline is depicted in Figure 1. In the following subsections, we first propose a novel overlapping fragmentation method and derive the fragment graph. Next, we introduce our model architecture, which takes fragment graph as input and produces the representation for both fragments and the whole molecule. Finally, we describe the nested masked fragment prediction pre-training, which aims to capture the hierarchical nature of fragments and their contextual relationships.

### 3.1 $k$-Degree Overlapping Fragmentation (DOVE)

Existing fragmentation methods mainly focus on non-overlapping fragmentation, which partition the molecule into disjoint atom sets (Lewell et al., 1998; Degen et al., 2008; Kong et al., 2022). The non-overlapping fragmentation methods have two drawbacks. First, one atom can only belong to one fragment, violating the fact that a single atom can be part of multiple functional groups (Merlot et al., 2003). Second, the induced fragment graph loses certain topological information from the original molecular graph. Such fragment graph cannot capture how two fragments are connected (see Appendix B for an illustration). To solve the above issues, we propose a novel method of overlapping fragmentation with learnable fragment library. It permits a single atom to participate in multiple fragments. The overlapping atoms can retain pivotal connection details between fragments, allowing for effective assembly and reducing connection information loss during the transformation from molecular graph to fragment graph. Our method is based on the iterative

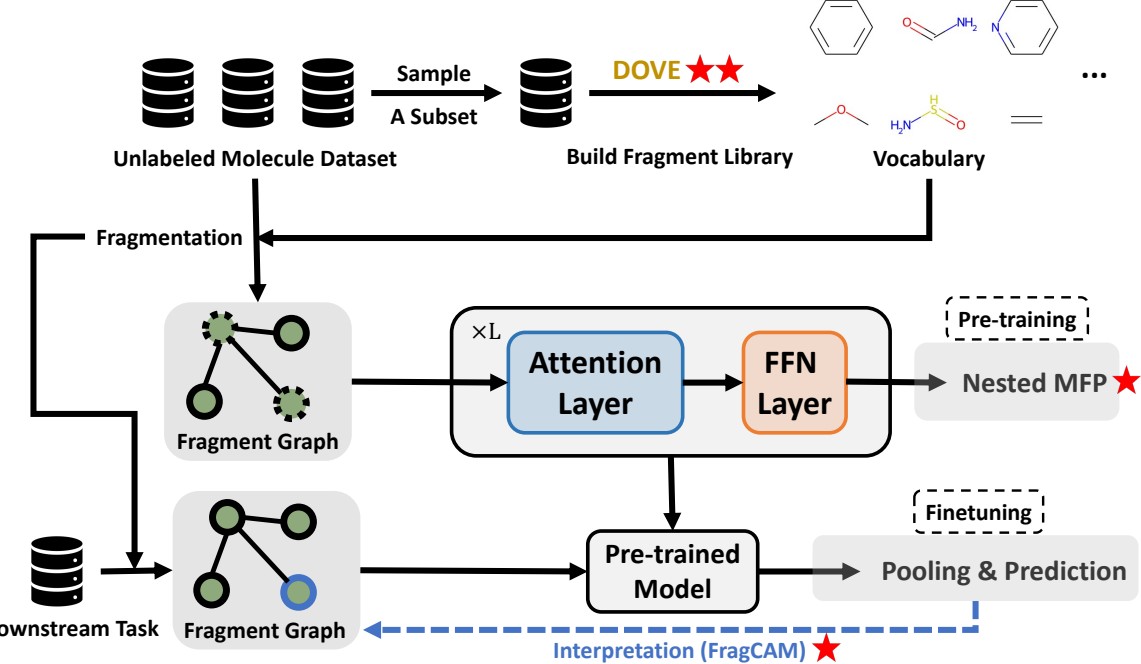

Figure 1: Pipeline of the FragFormer. "FFN": feed forward network. "MFP": masked fragment prediction. "★" represents our core design in FragFormer. We propose $k$-degree overlapping fragmentation (DOVE) to generate overlapping fragment graph, and nested MFP self-supervised learning to capture the hierarchical nature of fragments. FragCAM can effectively provide decisive fragments for prediction results.

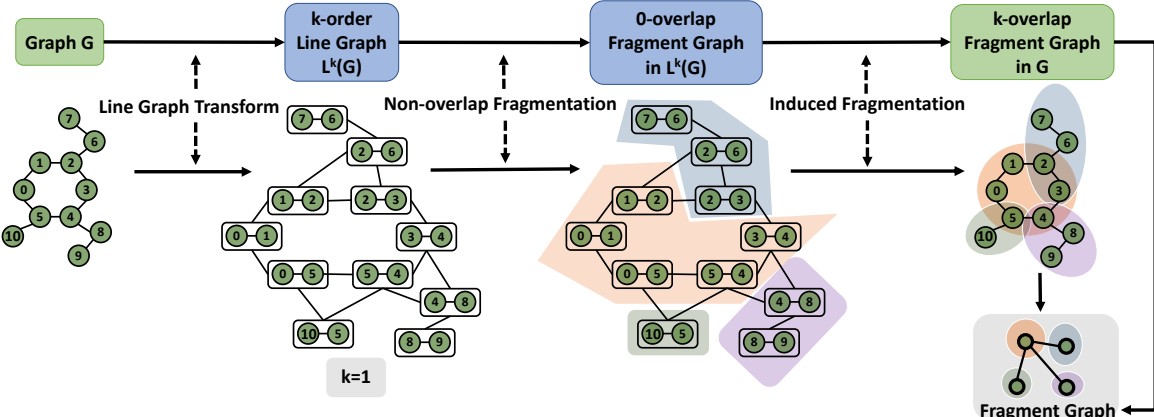

Figure 2: Pipeline of the $k$-degree overlapping fragmentation (DOVE) with $k = 1$.

line graph transformation of the molecular graph and can take advantage of the existing non-overlapping fragmentation methods. The pipeline is shown in Figure 2. The main idea is that *the non-overlapping fragmentation of the k-order line graph can be transformed to the k-degree overlapping fragmentation of the original molecular graph*. We first formally define $k$-degree overlapping fragmentation and $k$-order line graph. Then, we show that the $k$-degree overlapping fragmentation of the molecular graph can be obtained by the non-overlapping fragmentation of the $k$-order line graph. We assume all the graphs $G$ are undirected in the following description.

**Definition 1** (Fragmentation of graph). Given a graph $G = (V, E)$, $P = \{V_i\}_{i=1}^m$, $\mathbf{A} \in \mathbb{R}^{m \times m}$, we define $F = (P, A)$ as a fragmentation (fragment graph) of $G$ if

1. $P$ is a vertex cover of $G$, i.e, $\bigcup_{i=1}^{m} V_i = V$. $P$ also represents the nodes in fragment graph.
2. $\forall i \in \{1, 2, \cdots, m\}$, $G[V_i]$ is a connected subgraph. $G[V_i]$ represents the subgraph restricted to vertex set $V_i$.

Here, $\mathbf{A}$ symbolizes the adjacency matrix for the fragment graph, indicating whether two fragments are neighbors. This adjacency is flexible and can be adjusted based on specific requirements.

**Definition 2** ($k$-degree overlapping fragmentation)**.** Given a graph $G = (V, E)$, $P = \{V_i\}_{i=1}^{m}$, $\mathbf{A} \in \mathbb{R}^{m \times m}$, we call a fragmentation $F = (P, A)$ as a $k$-degree overlapping fragmentation if:

- $\forall\, i, j \in \{1, 2, \cdots, m\}, i \neq j$, if $A_{ij} = 1$, i.e., $V_i$ and $V_j$ are neighbors in $G$, then $|V_i \cap V_j| \geq k$.

Next, we define $k$-order line graph and show how it can be used to generate the $k$-degree overlapping fragmentation of the original molecular graph.

**Definition 3** (Line graph transformation)**.** Given a graph $G = (V, E)$, the line graph $L(G)$ is an undirected graph whose vertices correspond to the edges of $G$ and two vertices are connected by an edge if the corresponding edges in $G$ share a common vertex.

We assume $G$ is connected and leave the discussion for disconnected graph in Appendix A.

**Definition 4** ($k$-order line graph)**.** Let $L^0(G) = G$. For integer $k \geq 1$, the $k$-order line graph $L^k(G)$ is defined recursively as $L^k(G) = L(L^{k-1}(G))$.

Each vertex $v$ in $L^k(G)$ corresponds to a $(k+1)$-size connected subgraph in $G$. We denote its node set by $C(v)$.

**Definition 5** (Standard 0-degree overlapping fragmentation)**.** Given a graph $G = (V, E)$, $P = \{V_i\}_{i=1}^{m}$, $\mathbf{A} \in \mathbb{R}^{m \times m}$, we call a fragmentation $F = (P, A)$ as a standard 0-degree overlapping fragmentation if:

1. $\forall\, i, j \in \{1, 2, \cdots, m\}, i \neq j$, we have $V_i$ and $V_j$ are disjoint, i.e, $|V_i \cap V_j| = 0$.
2. $A_{ij} = 1$ if there exists node $v_i \in V_i$ and $v_j \in V_j$, such that $v_i$ and $v_j$ are neighbors in $G$, otherwise $A_{ij} = 0$.

Most commonly used fragmentation methods are standard 0-degree overlapping fragmentation, such as principal subgraph mining (Kong et al., 2022), BRICS (Degen et al., 2008) and Recap (Lewell et al., 1998).

**Definition 6** (Induced fragmentation)**.** Given a graph $G = (V, E)$ and a fragmentation $F = (P, A)$ of $L^k(G)$, where $P = \{U_i\}_{i=1}^{m}$, $\mathbf{A} \in \mathbb{R}^{m \times m}$, the induced fragmentation of $F$ is defined as $F' = (Q, A)$, where $Q = \{V_i\}_{i=1}^{m}$, $V_i = \bigcup_{v \in U_i} C(v)$.

**Theorem 1.** *Given a graph $G = (V, E)$, the induced fragmentation of standard 0-degree overlapping fragmentation of $L^k(G)$ is a $k$-degree overlapping fragmentation of $G$.*

The proof can be found in Appendix A. By Theorem 1, we first transform the molecular graph to the $k$-order line graph, then use existing graph fragmentation method to get the standard 0-degree overlapping fragmentation of $L^k(G)$, and finally get the $k$-degree overlapping fragmentation of $G$ by the induced fragmentation. The overlapping degree can be controlled by adjusting the order $k$ of the line graph. We apply principal subgraph mining (Kong et al., 2022) as our standard 0-degree overlapping fragmentation method, which can generate a succinct and fruitful fragment vocabulary.

## 3.2  Model Architecture of FragFormer

Assume the fragmentation of the molecular graph is $F = (P, \mathbf{A})$, where $P = \{V_i\}_{i=1}^{m}$ is the fragment set and $\mathbf{A}$ is the fragment adjacent matrix. The architecture of FragFormer is designed to model the contextual relationships among fragments and to learn the representation of those fragments. The model architecture of FragFormer consists of three main components: subgraph encoder, fragment-level graph transformer and knowledge-aware fusion. The overall architecture is shown in Figure 3.

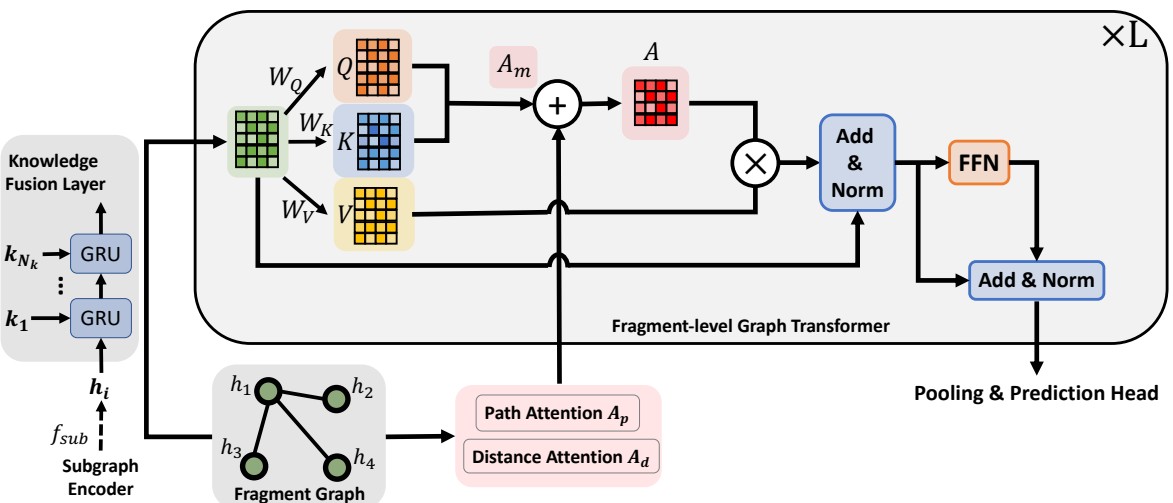

Figure 3: Model architecture of FragFormer.

**Subgraph Encoder**

Given a fragment $V_i$ in the overlapping fragmentation, we first obtain the embedding $\mathbf{h}_i$ of fragment $V_i$ by using a subgraph encoder $f_{sub}$:

$$\mathbf{h}_i = f_{sub}(G[V_i]).$$

$\{\mathbf{h}_i\}_{i=1}^m$ serve as the node features in the fragment graph.

**Fragment-level Graph Transformer**

Using fragment embeddings $\{\mathbf{h}_i\}_{i=1}^m$ and $\mathbf{A}$, we apply a transformer-based architecture (Vaswani et al., 2017) that operates on the fragment graph to model contextual dependency between fragments and learn a graph-level representation for molecular property prediction.

Fragment-level graph transformer is composed of $L$ graph transformer layers, each of which consists of a self-attention mechanism and a feed-forward module. We follow the design of self-attention mechanism in Li et al. (2023); Ying et al. (2021); Shi et al. (2022), which includes path attention and distance attention as a reflection of graph topology.

Specifically, given the fragment graph and fragment features $\mathbf{H}^l \in \mathbb{R}^{d \times m}$ at the $l$-th layer, where $d$ is the dimension of the feature and $m$ is the number of fragments, the self attention is calculated as:

$$\mathbf{Q} = \mathbf{W^Q}\mathbf{H}^l, \ \mathbf{K} = \mathbf{W^K}\mathbf{H}^l, \ \mathbf{V} = \mathbf{W^V}\mathbf{H}^l,$$

$$\mathbf{A} = \mathrm{softmax}\left(\frac{\mathbf{K^T}\mathbf{Q}}{\sqrt{d}} + \mathbf{A^d} + \mathbf{A^P}\right),$$

$$\mathbf{H}^{l+\frac{1}{2}} = \mathrm{LayerNorm}\left(\mathbf{VA} + \mathbf{H}^l\right),$$

$\mathbf{W^Q}$, $\mathbf{W^K}$ and $\mathbf{W^V} \in \mathbb{R}^{d \times d}$ are projection weight matrices for query ($\mathbf{Q}$), key ($\mathbf{K}$) and value matrix $\mathbf{V}$.

$\mathbf{A^d}$ and $\mathbf{A^P}$ are distance attention and path attention. Assume the distance between fragment $i$ and fragment $j$ in fragment graph is $N_p$, and the shortest path between fragment $i$ and fragment $j$ is $p_{ij} = (i_0, \ldots, i_{N_p})$, then $\mathbf{A^d}$ and $\mathbf{A^P}$ are computed as:

$$A_{ij}^d = c_{N_p},$$

$$A_{ij}^p = \frac{1}{N_p}\sum_{k=0}^{N_p}\mathbf{W}_k^p\mathbf{h}_{i_k},$$

where $c_{N_p}$ is a learnable scalar, $\mathbf{W}_k^p \in \mathbb{R}^{d \times 1}$ is a learnable weight matrix. The feed forward module is defined as:

$$\mathbf{H}^{l+1} = \text{LayerNorm}\left(\text{FFN}(\mathbf{H}^{l+\frac{1}{2}}) + \mathbf{H}^{l+\frac{1}{2}}\right),$$

where FFN is a two-layer MLP with ReLU activation (Agarap, 2018). The initial feature $\mathbf{H}^0$ is the fragment embedding $\{\mathbf{h}_i\}_{i=1}^m$. The output feature $\mathbf{H}^L$ is the final fragment representation. At last, we apply a global pooling operation on $\mathbf{H}^L$ to obtain the graph-level representation $\mathbf{h}_G$.

### Knowledge Fusion Layer

Incorporating fixed substructure-related features, such as extended connectivity fingerprints (ECFP) (Rogers & Hahn, 2010), are beneficial for molecular representation learning (Li et al., 2023; Rong et al., 2020). We refer to those fixed features as knowledge vectors. Here, we propose a knowledge fusion layer to incorporate the knowledge vectors into the fragment representation learning. We integrate the knowledge vector $\mathbf{k}$ with the fragment representation $\mathbf{h}_i$ through a GRU unit (Xiong et al., 2020) right after the subgraph encoder:

$$\mathbf{h}_i \leftarrow \text{GRU}\left(\mathbf{k}, \mathbf{h}_i\right).$$

If there are multiple knowledge vectors $\{\mathbf{k}_i\}_{i=1}^{N_k}$, we sequentially integrate them with the fragment representation, as illustrated in the bottom left corner of Figure 3.

### 3.3  Nested Masked Fragment Prediction Pre-training

To distill knowledge from massive unlabeled molecular data, it is essential to pre-train the model with a self-supervised objective. Here, we propose a novel fragment-based pre-training task, which is designed to model the hierarchical nature of fragments. Several atoms can assemble into small fragments, and a combination of these small fragments can constitute larger ones. Our design follows the philosophy of mask autoencoder (Devlin et al., 2019; He et al., 2022), where we mask the fragment nodes and predict the masked nodes based on the context. Instead of barely predicting the class label of the masked nodes, we predict the existence of all substructures in the vocabulary $V$ for the masked nodes. We use a 0-1 vector $\mathbf{y} \in \mathbb{R}^{|V|}$ to represent the existence of the substructure in the vocabulary. $y_i = 1$ if the $i$-th substructure exists in the masked node, otherwise $y_i = 0$. We predict the existence of the substructure by using a linear classifier built on the fragment representation $\mathbf{h}^L$ and a binary cross entropy loss:

$$\mathcal{L} = -\sum_{i=1}^{|V|} y_i \log\left(\text{sigmoid}\left(\mathbf{w}_i^T \mathbf{h}^L\right)\right) + (1 - y_i)\log\left(1 - \text{sigmoid}\left(\mathbf{w}_i^T \mathbf{h}^L\right)\right).$$

### 3.4  Fragment-level Interpretation: FragCAM

Inspired by the class activation mapping (Zhou et al., 2016) in CV, we combine it with our fragment modeling and propose **Frag**ment **C**lass **A**ctivation **M**apping (FragCAM) for explaining the prediction results with fragments. Assume the final fragment representation is $\left\{\mathbf{h}_i^L\right\}_{i=1}^m$ and we predict the molecular property with a linear classifier $\mathbf{w} \in \mathbb{R}^d$:

$$\hat{y} = \mathbf{w}^T \text{SumPooling}\left(\left\{\mathbf{h}_i^L\right\}_{i=1}^m\right).$$

Then, the attribution of the $i$-th fragment to the prediction result is calculated as:

$$\text{Attribution}_i = \frac{\mathbf{w}^T \mathbf{h}_i^L - \text{Min}}{\text{Max} - \text{Min}},$$

where Max and Min are the maximum and minimum value of $\left\{\mathbf{w}^T \mathbf{h}_i^L\right\}_{i=1}^m$, respectively.

## 4  Experiments

### 4.1  Pre-training

**Dataset**  We use two millions unlabeled molecular SMILES (Weininger, 1988) from CHEMBL29 (Gaulton et al., 2017) as our pre-training data. We use RDKit (Landrum, 2016) to generate the molecular graph from

Table 1: Results on PharmaBench. The best results are in bold. "↑" means the higher the better, while "↓" means the lower the better. FragFormer can achieve the SOTA preformance on eight out of eleven tasks. "‡" indicates traditional QSAR method. "∗" means the methods include a pre-training stage. Baselines labeled with "†" are fragment-based methods. The best performance of previous baseline methods (annotated with underlines) on each dataset mostly comes from methods based on atomic graphs.

| | Classfication (AUROC ↑) | | Regression (RMSE ↓) | | | | | | | | |
|---|---|---|---|---|---|---|---|---|---|---|---|
| Model/Dataset | AMES | BBB | CYP2C9 | CYP2D6 | CYP3A4 | HLMC | MLMC | RLMC | LogD | PPB | Sol |
| RF‡ (Breiman, 2001) | 0.761 | 0.731 | 18.471 | 18.041 | 16.540 | 0.813 | 0.987 | 0.958 | 1.249 | 0.204 | 0.918 |
| XGBoost‡ (Chen & Guestrin, 2016) | 0.768 | 0.750 | 17.582 | 17.819 | 16.123 | 0.647 | 0.844 | 0.819 | 1.071 | 0.186 | 0.832 |
| CMPNN (Song et al., 2020) | 0.858 | 0.887 | 18.377 | 19.156 | 16.701 | 0.921 | 1.130 | 0.939 | 0.807 | 0.236 | 0.858 |
| FPGNN (Cai et al., 2022) | 0.858 | 0.923 | 16.933 | 17.611 | 15.606 | 0.604 | 0.774 | 0.716 | 0.838 | 0.179 | 0.747 |
| DHTNN (Song et al., 2023) | 0.844 | 0.909 | 17.449 | 17.890 | 16.156 | 0.729 | 0.926 | 0.915 | 0.912 | 0.235 | 0.828 |
| KANO∗ (Fang et al., 2023) | 0.865 | 0.915 | 17.350 | 17.622 | 15.307 | 0.554 | 0.767 | 0.762 | 0.766 | 0.185 | 0.772 |
| MPG∗ (Li et al., 2021) | 0.869 | 0.923 | 17.417 | 17.527 | **14.376** | 0.541 | 0.723 | 0.685 | 0.758 | 0.170 | 0.758 |
| Unimol∗ (Zhou et al., 2023) | 0.878 | 0.920 | 17.774 | 18.071 | 15.895 | 0.613 | 0.824 | 0.651 | 0.745 | 0.179 | **0.707** |
| Trans-M∗ (Luo et al., 2023) | 0.869 | **0.935** | 18.080 | 17.677 | 15.867 | 0.567 | 0.744 | 0.677 | 0.737 | 0.172 | 0.834 |
| KPGT∗ (Li et al., 2023) | 0.880 | **0.935** | 17.036 | 16.860 | 16.379 | 0.564 | 0.726 | 0.881 | 0.728 | 0.172 | 1.221 |
| GraphFP∗† (Luong & Singh, 2023) | 0.830 | 0.893 | 17.367 | 21.183 | 17.219 | 0.764 | 0.878 | 0.771 | 0.835 | 0.208 | 1.935 |
| FraGAT† (Zhang et al., 2021b) | 0.778 | 0.684 | 17.788 | 22.503 | 20.313 | 0.775 | 0.849 | 1.050 | 0.945 | 0.220 | 1.352 |
| PharmHGT† (Jiang et al., 2023) | 0.863 | 0.913 | 17.490 | 15.020 | 16.077 | 0.544 | 0.820 | 0.677 | 0.676 | 0.172 | 0.954 |
| FragFormer-0∗† (ours) | 0.868 | 0.920 | 18.094 | 17.096 | 17.036 | 0.624 | 0.875 | 0.619 | 0.774 | 0.190 | 0.968 |
| FragFormer-1∗† (ours) | **0.889** | 0.928 | **16.855** | **14.425** | 15.894 | **0.514** | 0.702 | 0.596 | **0.667** | **0.157** | 0.895 |
| FragFormer-2∗† (ours) | 0.870 | 0.921 | 17.017 | 14.505 | 15.708 | 0.525 | **0.688** | **0.541** | 0.670 | 0.177 | 0.901 |
| FragFormer-3∗† (ours) | 0.870 | 0.918 | 17.835 | 16.780 | 16.977 | 0.545 | 0.732 | 0.571 | 0.714 | 0.193 | 0.961 |

SMILES and extract the atom features with 137 dimension. The detailed description of atom featurization can be found in Appendix D. We randomly sample $100k$ molecules to generate the fragment vocabulary by principal subgraph mining (Kong et al., 2022). We set the vocabulary size as 500 for the vocabulary constructon. We separately generate the library for different degree of overlapping fragmentation. The higher the degree of overlapping, the slower the vocabulary construction and molecule fragmentation. We select the degree of overlapping from $\{0, 1, 2, 3\}$ in our experiments. FragFormer with $k$-degree overlapping fragmentation is denoted as FragFormer-$k$. For the knowledge fusion layer, we extract ECFP (Rogers & Hahn, 2010) with radius $= 2$ and 1024 bits, MACCS (Durant et al., 2002), TorsionFP (Schulz-Gasch et al., 2012), and physic-chemical descriptors (Yang et al., 2019; Xue & Bajorath, 2000) as the knowledge vectors. We linearly transform the raw knowledge vectors to the same dimension as the fragment representation with a learnable projection for each kind of knowledge before the fusion.

**Training Details** We use $L = 6$ graph transformer layers in FragFormer with model dimension $d = 512$. We use a two-layer Graph IsoMorphism Network (GIN) (Xu et al., 2019) with mean pooling and hidden dimension $d$ as the subgraph encoder $f_{sub}$. We use batch size $bs = 4096$, learning rate $lr = 2e - 4$, and Adam optimizer (Kingma & Ba, 2015) with $(\beta_1, \beta_2) = (0.9, 0.999)$ in the stochastic training. We apply linear learning rate decay scheduler with $5k$ warm-up steps. We totally train the model for $25k$ steps on 4 NVIDIA 4090 GPUs. The masked fragment ratio is set as 0.3. The attention dropout rate and feed forward dropout rate are set as 0.1 in all fragment-level graph transformer layer. We adopt the multi-head attention technique with 8 heads in each layer (Vaswani et al., 2017). In the fine-tuning stage, we add a sum pooling on top of the pre-trained model and apply a linear predictor at last. More detailed description can be found in Appendix D.

## 4.2 Downstream Tasks

We test our method on molecular ADMET (Absorption, Distribution, Metabolism, Excretion, and Toxicity) property prediction benchmark PharmaBench (Niu et al., 2024), long-range biological benchmark (Dwivedi et al., 2022b), and interpretability benchmark DrugXAI for molecular property prediction (Rao et al., 2021).

**PharmaBench** PharmaBench (Niu et al., 2024) is a comprehensive benchmark for predicting molecular ADMET properties, featuring eleven distinct molecular property prediction tasks curated with the assistance of a multi-agent large language model system. The benchmark encompasses a total of 52,482 entries drawn from 14,401 bioassays, including two classification tasks and nine regression tasks. The detailed profile of the tasks can be found in the Appendix D. We follow the same scaffold splitting setting (Yang et al., 2019)

in PharmaBench which divides the dataset into training and test sets with a ratio of 4:1. We compare our method with traditional QSAR methods, atom-based methods and fragment-based methods. For traditional QSAR methods, we use ECFP descriptors with radius 2 and 1024 bits. For classification and regression tasks, we use AUROC and root mean square error (RMSE) as evaluation metrics, respectively. We report the mean performance of three runs with different random seeds.

**Long-range Biological Benchmark**  To assess the model's generalization preformance on large molecules and its capacity to capture long-range interactions, we evaluate it using the peptides-func datatset from the Long Range Graph Benchmark (Dwivedi et al., 2022b)[2] which predicts 10 biological functions of peptide data, e.g., Antibacterial, Antiviral, cell-cell communication, and others. This benchmark is designed to test the model's ability to model long range interactions between atoms. We compare our method with both graph convolutional based methods (Kipf & Welling, 2017; Chen et al., 2020a; Hu et al., 2020; Bresson & Laurent, 2017), transformer-based methods (Kreuzer et al., 2021) and fragment-based methods (Luong & Singh, 2023). Due to the large GPU memory cost for large molecule, we reduce the model dimension $d$ to 32 in this benchmark. We report the average precision (AP) for the benchmark with three different runs.

**DrugXAI**  Apart from the performance on molecular property prediction, explainability is also crucial for the model to be deployed in real-world applications. We evaluate our interpretation method FragCAM on DrugXAI (Rao et al., 2021). We choose two synthetic datasets and one real-world dataset from DrugXAI. Detailed descriptions of the datasets can be found in the Appendix C. Two synthetic datasets predict the existence of 3MR and Benzene substructures in the molecule, respectively. The ground truth attribution fragment is the corresponding substructure. The real-world dataset aims at predicting the ames mutagenicity. We use the 46 substructure alerts given by Sushko et al. (2012) as ground truth fragments. Following DrugXAI, we use AUROC metric proposed by McCloskey et al. (2019) to measure the model preformance, which computes the macro average AUROC between the predicted attribution and the ground truth attribution across test molecules. To evaluate FragCAM, we convert the fragment attribution into atom attribution by assigning the attribution of the fragment to the atoms it contains. If one atom is present in multiple fragments, we assign it the maximum attribution from those fragments. We compare our method with both atom-based and subgraph-based attribution approaches. For the atom-based methods, we adhere to the baseline settings established in DrugXAI. We apply CAM (Zhou et al., 2016), GradCAM (Selvaraju et al., 2020), GradInput (Shrikumar et al., 2017), and IG (Sundararajan et al., 2017) to CMPNN (**?**), GraphSAGE (Hamilton et al., 2017a), GraphNet (Battaglia et al., 2018), and GAT (Velickovic et al., 2018). We report the best results among the four atom-based attribution methods for each model. For subgraph-based method, we use SubgraphX (Yuan et al., 2021) on a two-layer GCN (Kipf & Welling, 2017) with dimension $d$. In SubgraphX, we set the target number of fragment nodes $N_{min}$ from $\{3, 7, 10, 15\}$, use 20 rollouts in MCTS, and apply 20 steps in shapley value estimation. We also report the fidelity and sparsity score (Pope et al., 2019; Yuan et al., 2023) for fragment attribution. The fidelity metric evaluates how accurately the explanations reflect the model's decision-making process. It removes the crucial substructures from the input graphs and analyze the variation in predictions. The sparsity metric quantifies the proportion of structures ignored by the explanation methods. Higher fidelity under similar or lower sparsity are preferred.

### 4.3  Results and Discussions

**PharmaBench**  The model performance on PharmaBench is presented in Table 1. Firstly, it is noteworthy that FragFormer achieves the SOTA performance in eight out of eleven tasks. Secondly, it consistently outperforms fragment-based baselines (indicated with †) with clear margins across all tasks. Thirdly, FragFormer employing overlapping fragmentation ($k = 1$ or $k = 2$) exhibits substantially superior performance in contrast to the non-overlapping fragmentation ($k = 0$). Excessive overlapping ($k = 3$) lead to performance degradation. We recommend using $k = 1$ or $k = 2$ for optimal results. These outcomes substantiate the efficiency of the FragFormer framework, particularly in context to its overlapping fragmentation feature. However, our method still falls short of atom-based pre-training methods on three tasks, which we aim to address in future work. We also find that for each task, the best method always involves a pre-training stage, which highlights the importance of pre-training in molecular property prediction.

---

[2]There is another peptide dataset called peptides-struct in the benchmark. However, peptides-struct pays attention to predicting the geometric quantities of peptides, which is beyond the scope of our work.

Table 2: Results on long-range biological benchmark. "w.o KF": without knowledge fusion layer.

| Model | # Params. | Test AP (↑) |
|---|---|---|
| GCN (Kipf & Welling, 2017) | 508k | 0.5930±0.0023 |
| GCNII (Chen et al., 2020a) | 505k | 0.5543±0.0078 |
| GINE (Hu et al., 2020) | 476k | 0.5498±0.0079 |
| GatedGCN (Bresson & Laurent, 2017) | 509k | 0.5864±0.0077 |
| GatedGCN (Bresson & Laurent, 2017)+RWSE (Dwivedi et al., 2022a) | 506k | 0.6069±0.0035 |
| GraphFP (Luong & Singh, 2023) | 2.5M | 0.6267±0.0073 |
| Transformer (Vaswani et al., 2017)+LapPE (Dwivedi et al., 2020) | 488k | 0.6326±0.0126 |
| SAN (Kreuzer et al., 2021)+LapPE (Dwivedi et al., 2020) | 493k | 0.6384±0.0121 |
| SAN (Kreuzer et al., 2021)+RWSE (Dwivedi et al., 2022a) | 500k | 0.6439±0.0075 |
| FragFormer (w.o KF) | 500k | **0.6571±0.0104** |
| FragFormer | 2M | **0.6693±0.0154** |

Table 3: Performance (AUROC) results on DrugXAI. The best results are in bold. For CMPNN, GraphSAGE, GraphNet, and GAT, we report the best results among CAM, GradCAM, GradInput, and IG.

| Dataset/Model | CMPNN | GraphSAGE | GraphNet | GAT | FragCAM |
|---|---|---|---|---|---|
| 3MR | 0.966 | 0.932 | 0.967 | 0.905 | **0.981** |
| Benzene | 0.906 | 0.859 | 0.934 | 0.867 | **0.935** |
| Mutagenicity | 0.742 | 0.742 | 0.759 | 0.683 | **0.793** |

**Long-range Biological Benchmark**  We show the results on the long-range biological benchmark in Table 2. FragFormer without knowledge fusion layer has the same amount of parameters as the baselines and outperforms the best baseline by more than 1% in terms of test AP. Compared to another fragment-based method GraphFP, FragFormer contains less parameters, but achieves a absolute improvement of 4.26% in test AP. These results demonstrate the effectiveness of FragFormer in processing large molecule and modeling long-range interactions in biological data.

**Interpretability Benchmark DrugXAI**  The results on DrugXAI are displayed in Table 3 and Table 4. In Table 3, FragCAM can significantly outperform the atom-based methods for baseline models on 3MR and mutagenicity dataset. For Benzene, FragCAM can also achieve competitive performance among all models. Compared to SubgraphX, FragCAM demonstrates superior AUROC and fidelity while achieving significantly faster speeds on the mutagenicity dataset. We also visualize the FragCAM results on mutagenicity datasets in Figure 4. FragCAM can accurately locate the nitro group, diazene and epoxyethane group as alerts for ames mutagenicity. These results demonstrate the effectiveness of FragCAM in explaining the prediction results with fragments.

### 4.4   Ablation Study

In this section, we use the PharmaBench dataset to analyze the impact of different module designs on the model performance. We set $k = 1$ in $k$-degree overlapping fragmentation for fair comparison. We define the *normalized performance* as the RMSE divided by the standard deviation of the labels for regression tasks. For classification tasks, it is calcuated by $1 - \mathrm{AUROC}$. A lower normalized value indicates better performance.

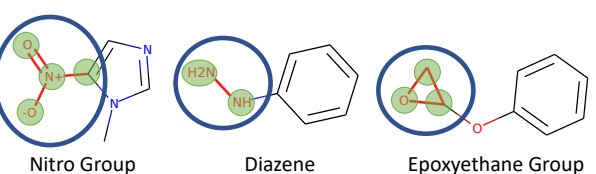

Nitro Group        Diazene        Epoxyethane Group

Figure 4: Interpretation results of FragCAM on mutagenicity datasets. The green nodes are the ground truth alerts. The blue fragments are the fragments with the highest attribution score in FragCAM.

Table 4: Performance comparison between FragCAM and SubgraphX on DrugXAI. The best results are in bold. $N_{min}$ represents the target number of fragment nodes in SubgraphX.

|  | SubgraphX | | | | FragCAM |
|---|---|---|---|---|---|
| $N_{min}$ | 3 | 7 | 10 | 15 | – |
| AUROC | 0.655 | 0.683 | 0.663 | 0.592 | **0.793** |
| Fidelity(sparsity) | 0.24(0.81) | 0.29(0.58) | 0.30(0.41) | 0.28(0.21) | **0.35(0.72)** |
| Time per sample (s) | 52 | 28 | 17 | 11 | **0.024** |

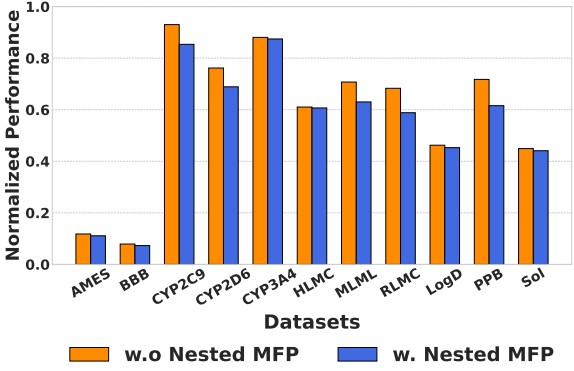 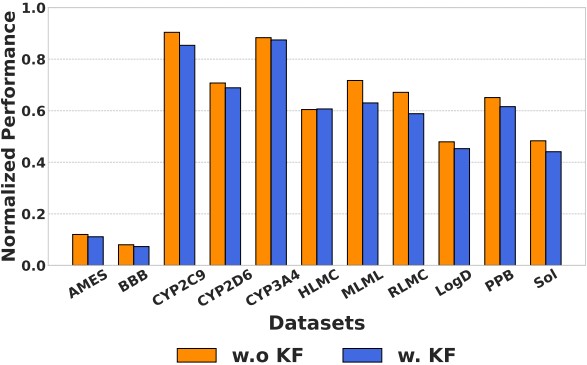

Figure 5: Left: Comparison between nested masked fragment prediction (w.) and cross entropy loss (w.o). "MFP": masked fragment prediction. Right: Comparison between models w./w.o knowledge fusion (KF).

Table 6: Comparison between different mask rates in pre-training.

| | Classfication (AUROC ↑) | | Regression (RMSE ↓) | | | | | | | | | Average Rank |
|---|---|---|---|---|---|---|---|---|---|---|---|---|
| Mask Rate | AMES | BBB | CYP2C9 | CYP2D6 | CYP3A4 | HLMC | MLMC | RLMC | LogD | PPB | Sol | |
| 0.1 | 0.879 | 0.924 | 17.886 | 16.676 | 16.298 | 0.523 | 0.733 | **0.592** | 0.698 | 0.174 | 0.922 | 2.2 |
| 0.3 | **0.889** | **0.927** | **16.855** | **14.425** | **15.894** | **0.514** | **0.702** | 0.596 | **0.667** | **0.157** | **0.895** | 1.1 |
| 0.5 | 0.868 | 0.919 | 17.527 | 16.627 | 16.379 | 0.527 | 0.738 | 0.621 | 0.689 | 0.180 | 0.924 | 2.7 |

**Effectiveness of Nested Masked Fragment Predictions** To evaluate the effect of the nested masked fragment prediction criterion, we pre-train the model with cross entropy loss which only predicts the identity of the masked fragment. The finetuning procedure remains unchanged. The results are illustrated in Figure 5 (left). We can see that the nested masked fragment prediction consistently outperforms the cross-entropy loss, often by a substantial margin, particularly for CYP2C9, CYP2D6, MLMC, RLMC, and PPB datasets.

**Impact of Knowledge Fusion Layer** We compare models with and without knowledge fusion layer. The results, displayed in Figure 5 (right), reveal that the model utilizing knowledge fusion layer outperforms its counterpart on most datasets except for HLMC, demonstrating the efficacy of our knowledge fusion module.

**How Vocabulary Size Affects Performance?** To evaluate the influence of vocabulary size, we vary the fragment vocabulary size to 200, 500, and 1000. The results are presented in Table 5. Notably, each vocabulary size achieves the best performance on at least one dataset. On average, a vocabulary size of 500 yields the best overall performance w.r.t average rank. If the vocabulary size is too small, it may not be able to capture the diversity of fragments, whereas an overly large vocabulary can introduce redundant fragments.

**Effect of Mask Rate in Pre-training** We investigate the effect of the mask rate in pre-training by adjusting it to 0.1, 0.3, and 0.5. The results, shown in Table 6, indicate that a mask rate of 0.3 provides the best performance across nearly all datasets. A mask rate that is too high can make the pre-training task too

Table 5: Comparison between different fragment vocabulary sizes.

| Vocab Size | Classfication (AUROC ↑) | | Regression (RMSE ↓) | | | | | | | | | Average Rank |
|---|---|---|---|---|---|---|---|---|---|---|---|---|
| | AMES | BBB | CYP2C9 | CYP2D6 | CYP3A4 | HLMC | MLMC | RLMC | LogD | PPB | Sol | |
| 200 | 0.855 | 0.923 | 17.177 | 15.033 | 15.960 | **0.508** | 0.705 | 0.610 | 0.670 | 0.171 | 0.910 | 2.7 |
| 500 | **0.889** | **0.927** | 16.855 | **14.425** | 15.894 | 0.514 | 0.702 | 0.596 | **0.667** | **0.157** | **0.895** | 1.3 |
| 1000 | 0.878 | 0.922 | **16.678** | 15.234 | **15.697** | 0.535 | **0.675** | **0.549** | 0.679 | 0.170 | 0.897 | 2.0 |

hard for the model to learn, while one that is too low may make the pre-training process excessively easy, offering little benefit for developing useful fragment representations.

## 5 Conclusion

In this paper, we propose a self-supervised learning framework, *FragFormer*, designed for molecular property prediction that enhances both performance and model interpretability. FragFormer features a novel molecule fragmentation method *DOVE* which generates overlapping fragments through iterative line graph transformation. Additionally, we implement a *nested masked fragment prediction* task to help the model learn the hierarchical fragment representations, along with a knowledge fusion layer to integrate fixed molecular features. To facilitate interpretability, we propose a simple yet effective method called *FragCAM*, which identifies the critical fragments contributing to predictions. FragFormer achieves state-of-the-art performance on eight out of eleven molecular property prediction tasks on PharmaBench and surpasses strong baseline models on long-range biological benchmark. Furthermore, FragFormer demonstrates superior interpretability on both synthetic and real-world molecular datasets, effectively identifying decisive fragments.

**Acknowledgments**

This work was supported by Beijing Natural Science Foundation NO. 4252046.

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

## A    Proof of Theorem 1

Recall that $L^k(G)$ is the $k$-order line graph of $G$. Each vertex $v$ in $L^k(G)$ corresponds to a $(k+1)$-size connected subgraph in $G$ and we denote its node set by $C(v)$. We assume $G$ is connected with more than $k$ vertices.

**Lemma 1.** *Given a graph $G = (V, E)$, for every two adjacent nodes $u$, $v$ in $L^k(G)$, we have $|C(u) \cap C(v)| \geq k$.*

*Proof.* Since $u$ and $v$ are adjacent in $L^k(G)$, they share a vertex in $L^{k-1}(G)$, which means they share a $k$-size connected subgraph in $G$. Thus, $|C(u) \cap C(v)| \geq k$. $\qquad\square$

**Theorem.** *Given a graph $G = (V, E)$, the induced fragmentation of standard $0$-overlapping fragmentation of $L^k(G)$ is a $k$-overlapping fragmentation of $G$.*

*Proof.* Denote the standard $0$-overlapping fragmentation of $L^k(G)$ by $F = (P, A)$ and the induced fragmentation as $F' = (Q, A)$, where $P = \{U_i\}_{i=1}^m$, $\mathbf{A} \in \mathbb{R}^{m \times m}$, $Q = \{V_i\}_{i=1}^m$, $V_i = \bigcup\limits_{v \in U_i} C(v)$. We will prove that $F'$ is a $k$-overlapping fragmentation of $G$.

1. $\bigcup\limits_{i=1}^m V_i = \bigcup\limits_{i=1}^m C(U_i) = V$, so $Q$ is a vertex cover of $G$.

Since $G[U_i]$ is connected, and every two adjacent nodes in $L^k(G)$ share at least $k$ nodes in $G$, so $G[V_i]$ is also connected.

Thus, $F'$ is a fragmentation of $G$.

2. If $A_{ij} = 1$, then there exists $u_i \in U_i$ and $u_j \in U_j$, $u_i$ and $u_j$ are adjacent in $L^k(G)$. By Lemma 1, $|C(u_i) \cap C(u_j)| \geq k$. Thus,

$$|V_i \cap V_j| = \left| (\bigcup_{v \in U_i} C(v)) \cap (\bigcup_{v' \in U_j} C(v')) \right| = \left| \bigcup_{v \in U_i, v' \in U_j} C(v) \cap C(v') \right| \geq |C(u_i) \cap (u_j)| \geq k$$

Combining 1 and 2, we conclude $F'$ is a $k$-overlapping fragmentation of $G$.

$\qquad\square$

**Disconnected Graph**    For a disconnected graph, we can apply the above theorem to each connected component. If a connected component contains only one node, we keep it unchanged in the line graph transformation.

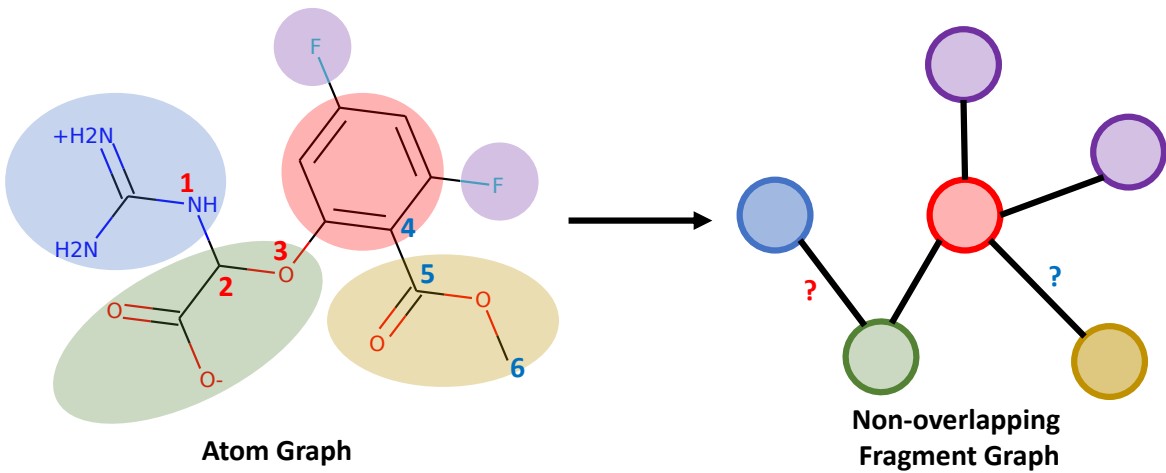

Figure 6: An example of connection information loss in non-overlapping fragmentation.

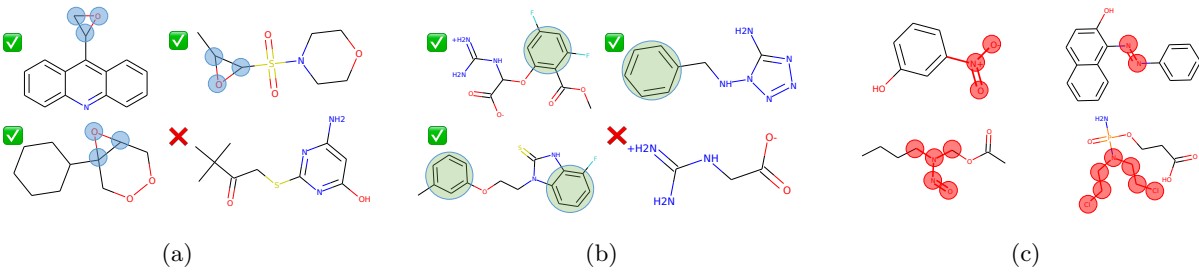

Figure 7: (a): 3MR dataset. The blue nodes indicate the 3MR substructure. The last molecule does not contain the 3MR substructure. (b) Benzene dataset. The green nodes indicate the benzene substructure. The last molecule does not contain the benzene substructure. (c) Mutagenicity dataset. The red nodes indicate the substructure alerts. All molecules are mutagenic in ames test.

## B   Connection Information Loss

We use an example to illustrate the connection information loss in the non-overlapping fragmentation. In Figure 6, we segment the molecule into six non-overlapping fragments and get the fragment graph. Within the fragment graph, the blue fragment is linked to the green fragment, but it's indistinguishable whether the connection is established through atom 1 and atom 2, or through atom 1 and atom 3. Similarly, the red fragment is connected with the yellow fragment, yet it remains unclear if this connection is formed by atom 4 and atom 5, or possibly atom 4 and atom 6. This specific kind of connection information is unfortunately lost within the context of a non-overlapping fragment graph. Based on our $k$-degree overlapping fragmentation, we can identify the connection information between fragments through the overlapping atoms. Since the connection information is preserved in the fragment graph, the model can access and *implicitly* utilize the connection information between fragments. We will explore more explicit ways to utilize the connection information in the future work.

## C   Interpretability Benchmark: DrugXAI

**3MR**   3MR dataset is a synthetic dataset that predicts the existence of 3MR substructure in the molecule. 3MR is a ring that consists of two carbon atoms and one oxygen atom, all connected by single bonds. The examples are given in Figure 7(a). The dataset is collected from ZINC15 lead-like subset and the ground truth attribution is the 3MR substructure.

Table 7: Dataset size of 3MR, Benzene and Mutagenicity.

|              | Train Set Size | Test Set Size |
| ------------ | -------------- | ------------- |
| 3MR          | 2521           | 631           |
| Benzene      | 9600           | 2400          |
| Mutagenicity | 5204           | 1302          |

Table 8: Performance (AUROC) of CMPNN, GraphSAGE, GraphNet, GAT with CAM, GradCAM, GradInput, IG on 3MR dataset.

| 3MR       | CMPNN     | GraphSAGE | GraphNet  | GAT       |
| --------- | --------- | --------- | --------- | --------- |
| CAM       | 0.858     | 0.771     | 0.832     | 0.734     |
| GradCAM   | 0.794     | 0.783     | 0.754     | 0.745     |
| GradInput | 0.951     | 0.844     | **0.967** | 0.877     |
| IG        | **0.966** | **0.932** | 0.942     | **0.905** |
| Best      | 0.966     | 0.932     | 0.967     | 0.905     |

**Benzene** Benzene dataset is a synthetic dataset designed to predict the presence of the benzene substructure within a molecule. Benzene is a ring consisting of six carbon atoms connected by aromatic bonds. The examples are illustrated in Figure 7(b). The dataset is also sourced from the ZINC15 lead-like subset, with the ground truth attribution being the benzene substructure.

**Mutagenicity** Mutagenicity dataset aims to predict the ames mutagenicity of molecules (Ames et al., 1973). We use the 46 substructure alerts from Sushko et al. (2012) as ground truth fragments.

The size of train and test set after dataset splitting are shown in Table 7. We report the test set attribution AUROC (McCloskey et al., 2019) of CMPNN, GraphSAGE, GraphNet, GAT with CAM, GradCAM, GradInput. in Table 8 (3MR dataset), Table 9 (Benzene datase), and Table 10 (Mutagenicity dataset). These results refer to DrugXAI (Rao et al., 2021). We list them here for the convenience of the reader.

**Other Alternatives of FragCAM** We adopt DeepLIFT (Shrikumar et al., 2017) and Integrated Gradients (IG) (Sundararajan et al., 2017) on the initial fragment vectors, which is the output of subgraph encoder, and denote these methods as FragDeepLIFT and FragIG. We set steps=50 for IG and use the default setting for DeepLIFT in Captum (Kokhlikyan et al., 2020). The results on DrugXAI are shown in Table 11 and Table 12. FragCAM outperforms FragDeepLIFT and FragIG on all datasets in terms of AUROC and fidelity, which demonstrates the effectiveness of FragCAM in explaining the prediction results with fragments.

## D Experimental Details

**Atom Features** The composition of atom features is shown in Table 13.

**Profile of PharmaBench** The profile of the PharmaBench dataset is shown in Table 14.

**Finetuning Details** In the fine-tuning stage, we add a sum pooling on top of the pre-trained model and apply a linear predictor at last. We use Adam optimizer with a learning rate of $3e-5$ and a batch size of 32. We train the model for 50 epochs and report the mean performance on the test set over three runs with different random seeds. The model is implemented in DGL (Wang et al., 2019) with PyTorch (Paszke et al., 2019) as backend and trained on a single NVIDIA A800 GPU.

**Time Measurement on Mutagenicity Dataset** We measure the time per sample of FragCAM and SubgraphX on the mutagenicity dataset on a single NVIDIA 4090 GPU.

Table 9: Performance (AUROC) of CMPNN, GraphSAGE, GraphNet, GAT with CAM, GradCAM, GradInput, IG on Benzene dataset.

| Benzene | CMPNN | GraphSAGE | GraphNet | GAT |
|---|---|---|---|---|
| CAM | 0.831 | 0.798 | 0.859 | 0.776 |
| GradCAM | 0.566 | 0.582 | 0.657 | 0.604 |
| GradInput | 0.894 | **0.859** | **0.934** | **0.867** |
| IG | **0.906** | 0.801 | 0.865 | 0.798 |
| Best | 0.906 | 0.859 | 0.934 | 0.867 |

Table 10: Performance (AUROC) of Mutagenicity, GraphSAGE, GraphNet, GAT with CAM, GradCAM, GradInput, IG on Benzene dataset.

| Mutagenicity | CMPNN | GraphSAGE | GraphNet | GAT |
|---|---|---|---|---|
| CAM | 0.539 | 0.542 | 0.539 | 0.494 |
| GradCAM | 0.560 | 0.542 | 0.539 | 0.503 |
| GradInput | 0.607 | 0.607 | 0.654 | 0.608 |
| IG | **0.742** | **0.742** | **0.759** | **0.683** |
| Best | 0.742 | 0.742 | 0.759 | 0.683 |

# E More Ablation Study on PharmaBench

**Comparison between Different Fragmentation Hierarchies** We first construct three hierarchy of fragmentation: All Atom, Junction Tree (JT) (Jin et al., 2018), and DOVE-1. "All Atom" takes each atom as a fragment. JT decomposes the molecule into rings and bonds. The number of fragments follows All Atom > JT > DOVE. We then evaluate the performance of FragFormer with different fragmentation methods, as shown in Table 15. The results show that the overall performance of FragFormer with DOVE is better than JT, which is better than All Atom.

**Comparison between DOVE and JT-VAE** JT-VAE (Jin et al., 2018) decomposes the molecule into rings and bonds. DOVE decomposes the molecule based on iterative line graph and principal subgraph mining (Kong et al., 2022). Compared with JT-VAE, DOVE can learn a succinct and expressive fragment library from molecule database by mining frequent substructures. The chemical functional groups in DOVE are more diverse and flexible than the rings and bonds in JT-VAE. Moreover, DOVE can decompose the molecule into fragments with tunable overlapping degree, which is not supported by JT-VAE. We constuct a baseline model called FragFormer-JT, which replace the DOVE with the Junction Tree fragmentation (JT-VAE) and keep all other settings the same. The results on PharmaBench are shown in Table 15. FragFormer-JT achieves lower performance than FragFormer on Pharmabench, which demonstrates the effectiveness of DOVE.

**Comparison between Different $k$ with the Same Vocabulary** Vocabulary construction inherently depends on k, as it is performed on $L^k(G)$. We conduct an ablation experiment to isolate the difference of vocabulary. We first build the vocabulary for $k = 0, 1, 2, 3$ with vocab_size=500 and merge the vocabularies to get the final vocabulary. Then, we pre-train the model for $k = 0, 1, 2, 3$ with the final vocabulary separately and evaluate the performance on Pharmabench. The results are shown in Table 16. The optimal results for each dataset occur at $k = 1$ or $k = 2$, which is consistent with the findings in Table 1. Interestingly, under the same vocabulary, $k = 2$ yields superior performance more frequently than $k = 1$. The trend is reversed in Table 1.

**Effect of Key Components Through an Incremental Ablation Study** We begin with the baseline model, which employs mask atom prediction without incorporating knowledge fusion ('FragFormer-Atom w/o K'). Next, we introduce DOVE fragmentation with $k = 1$ ('FragFormer w/o K & Nested MFP'). Subsequently,

Table 11: Performance (AUROC) of FragCAM, FragIG, and FragDeepLift on DrugXAI.

| Dataset/Method | FragCAM | FragIG | FragDeepLift |
|:---:|:---:|:---:|:---:|
| 3MR | **0.981** | 0.971 | 0.934 |
| Benzene | **0.935** | 0.886 | 0.875 |
| Mutagenicity | **0.793** | 0.711 | 0.704 |

Table 12: Fidelity(sparsity) of FragCAM, FragIG, and FragDeepLift on DrugXAI.

| Dataset/Method | FragCAM | FragIG | FragDeepLift |
|:---:|:---:|:---:|:---:|
| 3MR | **0.87(0.85)** | 0.87(0.78) | 0.13(0.75) |
| Benzene | **0.50(0.67)** | 0.46(0.64) | 0.45(0.64) |
| Mutagenicity | **0.35(0.72)** | 0.36(0.67) | 0.19(0.66) |

we incorporate the nested MFP pre-training criterion ('FragFormer w/o K'). Finally, we enhance the model by adding the knowledge fusion layer ('FragFormer'). The results are shown in Table 17. Each component contributes to the performance improvement.

## F    Results on MoleculeNet

We test FragFormer (DOVE-1) on the MoleculeNet benchmark (Wu et al., 2017). We follow the experimental settings in KPGT (Li et al., 2023). We split the dataset into 80% training, 10% validation, and 10% test with scaffold splitting. We report the mean performance over three runs with different random splits. The results are shown in Table 18. All baseline results of atom-based models are taken from KPGT. FragFormer outperforms all fragment-based baselines and achieves the SOTA performance on 7 out of 11 datasets. We find that FragFormer excels at the target-related tasks, e.g., BBBP, BACE, Estrogen, while underperforms previous SOTA method on toxicity-related tasks, e.g., Tox21, ToxCast. We hypothesize that this could be due to differences in task characteristics, and we plan to investigate the underlying reasons for this phenomenon in future work.

Table 13: Composition of atom features.

| Feature Description | Feature Dimension |
|---|---|
| One hot encoding for the atom type, e.g., C, N, O | 101 |
| One hot encoding for the atom degree | 12 |
| Formal charge | 1 |
| One hot encoding for radical electrons | 6 |
| One hot encoding for atom hybridization | 6 |
| Whether the atom is in aromatic ring | 1 |
| One hot encoding of number of H | 6 |
| Whether the atom is chiral center | 1 |
| One hot encoding of chirality type | 2 |
| normalized atom mass | 1 |

Table 14: Datasets Profile for PharmaBench.

| Dataset Name | Dataset Size | Task Type | Category | Task Description |
|---|---|---|---|---|
| AMES | 9,139 | classification | Toxicity | The result of AMES test (Ames et al., 1973) which evaluates the mutagenic potential of the molecule. |
| BBB | 8,301 | classification | Absorption | Predict the Blood-Brain Barrier (BBB) penetration. The dataset is labeled according to threshold $\log BB = -1$. |
| CYP2C9 | 999 | regression | Metabolism | Binding affinity to CYP2C9 (unit: $\log 10\mu M$). |
| CYP2D6 | 1,214 | regression | Metabolism | Binding affinity to CYP2D6 (unit: $\log 10\mu M$). |
| CYP3A4 | 1,980 | regression | Metabolism | Binding affinity to CYP3A4 (unit: $\log 10\mu M$). |
| HLMC | 2,286 | regression | Clearance | The clearance speed in the liver microsomal system in human (unit: $\log 10(\mathrm{mL/min/g})$). |
| MLMC | 1,403 | regression | Clearance | The clearance speed in the liver microsomal system in mouse (unit: $\log 10(\mathrm{mL/min/g})$). |
| RLMC | 1,129 | regression | Clearance | The clearance speed in the liver microsomal system in rat (unit: $\log 10(\mathrm{mL/min/g})$). |
| LogD | 13,068 | regression | Physochemical | LogD is the logarithm of the distribution coefficient (D), which measures PH-adjusted lipophilicity of the molecule. |
| PPB | 1,262 | regression | Distribution | Percentage of the molecule in the plasma that is bound. |
| Sol | 11,701 | regression | Physochemical | Water solubility of the molecule (unit: $\log 10 nM$). |

Table 15: Comparison between different hierarchy of fragmentation.

| | Classfication (AUROC ↑) | | Regression (RMSE ↓) | | | | | | | | |
|---|---|---|---|---|---|---|---|---|---|---|---|
| Method/Dataset | AMES | BBB | CYP2C9 | CYP2D6 | CYP3A4 | HLMC | MLMC | RLMC | LogD | PPB | Sol |
| FragFormer-Atom | 0.867 | 0.916 | 17.849 | 16.133 | 16.663 | 0.606 | 0.755 | 0.718 | 0.800 | 0.192 | 1.045 |
| FragFormer-JT | 0.863 | 0.919 | 17.413 | 15.666 | 16.603 | 0.550 | 0.746 | 0.711 | 0.709 | 0.185 | 0.938 |
| FragFormer-DOVE | **0.889** | **0.927** | **16.855** | **14.425** | **15.894** | **0.514** | **0.702** | **0.596** | **0.667** | **0.157** | **0.895** |

Table 16: Comparison between different overlapping degree $k$ with the same vocabulary.

| | Classfication (AUROC ↑) | | Regression (RMSE ↓) | | | | | | | | |
|---|---|---|---|---|---|---|---|---|---|---|---|
| Method/Dataset | AMES | BBB | CYP2C9 | CYP2D6 | CYP3A4 | HLMC | MLMC | RLMC | LogD | PPB | Sol |
| FragFormer-0 | 0.858 | 0.918 | 17.358 | 17.241 | 16.859 | 0.625 | 0.854 | 0.603 | 0.780 | 0.193 | 0.952 |
| FragFormer-1 | 0.865 | **0.922** | 17.316 | 14.833 | 15.773 | **0.517** | 0.734 | 0.565 | 0.695 | **0.175** | 0.910 |
| FragFormer-2 | **0.870** | 0.920 | **16.732** | **14.538** | **15.302** | 0.519 | **0.708** | **0.548** | **0.687** | 0.177 | **0.905** |
| FragFormer-3 | 0.860 | 0.917 | 17.809 | 16.986 | 16.910 | 0.544 | 0.754 | 0.577 | 0.725 | 0.199 | 0.959 |

Table 17: Effect of key components.

| Method/Dataset | Classfication (AUROC ↑) | | Regression (RMSE ↓) | | | | | | | | |
|---|---|---|---|---|---|---|---|---|---|---|---|
| | AMES | BBB | CYP2C9 | CYP2D6 | CYP3A4 | HLMC | MLMC | RLMC | LogD | PPB | Sol |
| FragFormer-Atom w.o K | 0.837 | 0.913 | 18.264 | 16.691 | 17.190 | 0.620 | 0.996 | 0.937 | 0.894 | 0.236 | 1.387 |
| FragFormer w.o (K&Nested MFP) | 0.856 | 0.921 | 17.919 | 15.659 | 16.894 | 0.563 | 0.877 | 0.774 | 0.740 | 0.211 | 1.134 |
| FragFormer w.o K | 0.882 | 0.921 | 17.851 | 14.815 | 16.055 | **0.512** | 0.799 | 0.680 | 0.706 | 0.166 | 0.981 |
| FragFormer | **0.889** | **0.928** | **16.855** | **14.425** | **15.894** | 0.514 | **0.702** | **0.596** | **0.667** | **0.157** | **0.895** |

Table 18: Results on MoleculeNet. Baselines labeled with "†" are fragment-based methods.

| Method | Classification (AUROC ↑) | | | | | | | | Regression (RMSE ↓) | | |
|---|---|---|---|---|---|---|---|---|---|---|---|
| | BACE | BBBP | ClinTox | SIDER | Estrogen | MetStab | Tox21 | ToxCast | FreeSolv | ESOL | Lipo |
| Infomax (Velickovic et al., 2019) | 0.839 | 0.840 | 0.661 | 0.616 | 0.888 | 0.837 | 0.816 | 0.690 | 4.119 | 1.462 | 0.978 |
| Edgepred (Hamilton et al., 2017b) | 0.817 | 0.873 | 0.730 | 0.603 | 0.881 | 0.844 | 0.818 | 0.712 | 3.849 | 2.272 | 1.030 |
| Masking (Hu et al., 2020) | 0.823 | 0.864 | 0.729 | 0.573 | 0.869 | 0.868 | 0.798 | 0.663 | 3.646 | 2.100 | 1.063 |
| Contextpred (Hu et al., 2020) | 0.840 | 0.877 | 0.732 | 0.609 | 0.882 | 0.857 | 0.806 | 0.714 | 3.141 | 1.349 | 0.969 |
| Infomax_sup (Hu et al., 2020) | 0.839 | 0.873 | 0.754 | 0.622 | 0.864 | 0.864 | 0.826 | 0.713 | 3.017 | 1.238 | 0.729 |
| Edgepred_sup (Hu et al., 2020) | 0.847 | 0.859 | 0.745 | 0.620 | 0.890 | 0.852 | 0.829 | 0.721 | 2.889 | 1.133 | 0.707 |
| Masking_sup (Hu et al., 2020) | 0.824 | 0.859 | 0.796 | 0.606 | 0.888 | 0.872 | 0.827 | 0.715 | 3.210 | 1.387 | 0.725 |
| Contextpred_sup (Hu et al., 2020) | 0.855 | 0.875 | 0.802 | 0.620 | 0.885 | 0.859 | 0.840 | 0.724 | 3.105 | 1.477 | 0.754 |
| GraphLoG (Xu et al., 2021) | 0.830 | 0.846 | 0.667 | 0.615 | 0.871 | 0.850 | 0.796 | 0.677 | 4.174 | 2.335 | 1.104 |
| GraphCL (You et al., 2020) | 0.825 | 0.887 | 0.691 | 0.587 | 0.875 | 0.821 | 0.805 | 0.696 | 4.014 | 1.835 | 0.945 |
| JOAO (You et al., 2021) | 0.826 | 0.879 | 0.741 | 0.640 | 0.861 | 0.837 | 0.823 | 0.711 | 3.466 | 1.771 | 0.933 |
| GROVER (Rong et al., 2020) | 0.840 | 0.887 | 0.874 | 0.638 | 0.892 | 0.876 | 0.838 | 0.696 | 2.991 | 0.928 | 0.752 |
| 3DInfomax (Stärk et al., 2022) | 0.811 | 0.877 | 0.887 | 0.585 | 0.880 | 0.866 | 0.805 | 0.716 | 2.919 | 1.906 | 1.045 |
| GraphMVP (Liu et al., 2022) | 0.818 | 0.860 | 0.719 | 0.584 | 0.865 | 0.820 | 0.799 | 0.689 | 2.532 | 1.937 | 0.990 |
| MolFormer (Ross et al., 2021) | 0.791 | 0.866 | 0.810 | 0.578 | 0.806 | 0.651 | 0.764 | 0.687 | 2.322 | 0.821 | 0.673 |
| ImageMol (Zeng et al., 2022) | 0.786 | 0.881 | 0.885 | 0.625 | 0.839 | 0.874 | 0.816 | 0.710 | 2.634 | 1.869 | 0.765 |
| GEM (Fang et al., 2021) | 0.857 | 0.895 | 0.905 | 0.621 | 0.894 | 0.863 | 0.832 | 0.733 | 2.389 | 0.803 | 0.663 |
| GraphMAE (Hou et al., 2022) | 0.857 | 0.878 | 0.748 | 0.597 | 0.881 | 0.853 | 0.801 | 0.691 | 3.023 | 1.378 | 0.746 |
| MoleBERT (Xia et al., 2023) | 0.843 | 0.851 | 0.797 | 0.615 | 0.887 | 0.868 | 0.832 | 0.720 | 2.801 | 1.185 | 0.690 |
| KPGT (Li et al., 2023) | 0.855 | 0.908 | **0.946** | 0.649 | 0.905 | 0.889 | **0.848** | **0.746** | 2.121 | 0.803 | **0.600** |
| GraphFP† (Luong & Singh, 2023) | 0.839 | 0.869 | 0.832 | 0.598 | 0.879 | 0.857 | 0.813 | 0.707 | 3.228 | 1.805 | 0.803 |
| FraGAT† (Zhang et al., 2021b) | 0.813 | 0.841 | 0.726 | 0.577 | 0.867 | 0.841 | 0.771 | 0.682 | 4.049 | 2.043 | 0.913 |
| PharmHGT† (Jiang et al., 2023) | 0.846 | 0.891 | 0.895 | 0.625 | 0.892 | 0.874 | 0.824 | 0.715 | 2.411 | 1.276 | 0.779 |
| FragFormer† | **0.868** | **0.912** | 0.919 | **0.656** | **0.918** | **0.894** | 0.840 | 0.732 | **1.990** | **0.801** | 0.642 |

