# OpenReview forum: "FragFormer: A Fragment-based Representation Learning Framework for Molecular Property Prediction"
_TMLR — Accepted by TMLR_

### Review · Reviewer_BMQK · 2025-01-10

**Summary Of Contributions:**

- FragFormer is a self-supervised learning framework for molecular representation. It leverage the prior that a molecule is consisted of overlapping molecular fragments.
- This work develops a new k-Degree Overlapping fragmentation (DOVE) method - to account for overlapping functional groups.
- The fragment-level interpretation method FragCAM is proposed and outperforms all baseline methods.
- The framework is proved effective for molecular prediction tasks - reaching state-of-the-art in 8 out of 11 tasks in PharmaBench.

**Audience:**

Yes

**Claims And Evidence:**

Yes

**Requested Changes:**

- It would be ideal to see a ablation and discussion on how the overlapping fragmentation method purposed in the method compare to the fragmentation method in JT-VAE. Is the newly proposed fragmentation method essential for FragFormer’s success?

- Clarification on whether the model can directly access and utilize the bonding information between fragments from DOVE. The claim that bonding information is preserved may need to be adjusted.

**Strengths And Weaknesses:**

Strengths:

- The paper is very well-written - related work sufficiently covers the background and motivates the work; The method section is concise and clear.
- The interpretability at fragment level for molecular property is highly useful contribution to the field; FragCAM report the best results among all baselines.
- Ablation results sufficiently motivate various design choices. Overlapping fragmentation exhibits substantially superior performance in contrast to the non-overlapping fragmentation - proving the effectiveness of DOVE.
- FragFormer outperforms exiting methods in most molecular property prediction tasks.

Weaknesses:
- One the main contribution of this work is proposing overlapping fragmentation of the molecular. However overlapping fragmentation has been purposed previously in JT-VAE in 2018. Bonding information is preserved via overlapping atoms in clusters. DOVE with degree 1- the best performing setting for molecular setting, seem to be produce fragment graph that is identical to JT-VAE’s method. There’s no discussion or ablation against the fragmentation method from JT-VAE in the paper.


- It is claimed that one of the advantage for the proposed fragmentation method is that it preserves bonding information in the fragment graph - because there’s overlapping atoms in a pair of connected fragment nodes. However, the model architecture cannot directly access the bonding information - since the Fragment-level Graph Transformer just accesses the fragment embedding as input - which is computed independently for each molecular subgraph. It doesn't seem possible for the model to recover the bonding information between fragments from the subgraph pair.

---

> ### Author Response · Authors · 2025-02-13
> **Review Response**
>
> We thank the reviewer for the encouraging and insightful comments. We have revised our manuscript according to these comments and suggestions. Please find our responses to specific questions and concerns below.
>
> **Question 1**: DOVE with degree 1 seems to be produce fragment graph that is identical to JT-VAE's method.
>
> **Response**: We thank the reviewer for pointing out the comparison between DOVE and JT-VAE.
> The two methods are fundamentally different. DOVE decomposes the molecule based on iterative line graph and principal subgraph mining[1], while JT-VAE[2] decomposes the molecule into rings and bonds. DOVE relies on statistical analysis, while JT-VAE adheres to human-defined rules. Following the philosophy of BPE in NLP, DOVE can learn a succinct and expressive fragment library from molecule database by mining frequent substructures. The chemical functional groups in DOVE are more diverse and flexible than the rings and bonds in JT-VAE. Moreover, DOVE can decompose the molecule into fragments with tunable overlapping degree, which is not supported by JT-VAE.
> We include the discussion and comparison in the revised manuscript (Appendix E) to eliminate the ambiguity.
>
> **Question 2**: It would be ideal to see a discussion or ablation against the fragmentation method from JT-VAE in the paper. Is the newly proposed fragmentation method essential for FragFormer's success?
>
> **Response**: We thank the reviewer for the constructive suggestion. We add a baseline model called FragFormer-JT, which replace the DOVE (k=1) with the Junction Tree fragmentation (JT-VAE) and keep all other settings the same. The results on PharmaBench show that FragFormer-JT achieves lower performance than FragFormer on Pharmabench, which demonstrates the effectiveness of DOVE. Compared to JT-VAE, DOVE can learn a more expressive fragment library from the molecule database and the chemical functional groups in DOVE are more diverse and flexible.
> We include the discussion and results in the revised manuscript (Appendix E).
>
> | Method / Dataset            |  AMES(CLS)  |  BBB(CLS)   |  CYP2C9  |  CYP2D6  |  CYP3A4  |  HLMC  |  MLMC  |  RLMC  |  LogD  |  PPB   |  Sol   |
> |--------------------|--------|--------|----------|----------|----------|--------|--------|--------|--------|--------|--------|
> | FragFormer-JT      | 0.863  | 0.919  | 17.413   | 15.666   | 16.603   | 0.550  | 0.746  | 0.711  | 0.709  | 0.185  | 0.938  |
> | FragFormer-DOVE    | **0.889** | **0.927** | **16.855** | **14.425** | **15.894** | **0.514** | **0.702** | **0.596** | **0.667** | **0.157** | **0.895** |
>
> (CLS represents a classification task, measured by AUROC, where higher values are better. The others are regression tasks, measured by RMSE, where lower values are better.)
> **Question 3**: Clarification on whether the model can directly access and utilize the bonding information between fragments from DOVE. The claim that bonding information is preserved may need to be adjusted.
>
> **Response**: We thank the reviewer for raising this important point. Our claim is that DOVE can preserve the connection information during the fragmentation phase compared to non-overlapping fragmentation. Since the bonding information is preserved in the fragment graph, the model can access and *implicitly* utilize the bonding information between fragments. We will explore more explicit ways to utilize the bonding information in the future work.
> We also clarify this ambiguity in the revised manuscript (last sentence in Appendix B).
>
> [1] Xiangzhe Kong et al., Molecule Generation by Principal Subgraph Mining and Assembling, NeurIPS 2022
> [2] Wengong Jin et al., Junction Tree Variational Autoencoder for Molecular Graph Generation, ICML 2018

---

> > ### Comment · Reviewer_BMQK · 2025-02-19
> >
> > Thank you for the clarifications and new experiments. This addresses all my questions.

---

### Review · Reviewer_BNJd · 2025-01-15

**Summary Of Contributions:**

The authors propose FragFormer, a framework for molecular property prediction. The approach consists of (1) k-Degree Overlapping Fragmentation and (2) nested masked fragment prediction. The authors also propose a interpretation technique, FragCAM, to improve the interpretability of molecular representation. The main focus is to fully utilize overlapping fragmentations, which is yet under-explored in previous fragment-based approaches. Experimental results show that FragFormer achieves decent performance in various molecular property prediction tasks.

**Audience:**

Yes

**Claims And Evidence:**

Yes

**Requested Changes:**

See Weaknesses and Questions.

**Strengths And Weaknesses:**

Strengths:

1. The problem of interest, molecular property prediction, is an important topic in chemistry, e.g., drug discovery.

2. The motivation is reasonable; incorporating hierarchical nature of fragments can improve the understanding of molecules.

3. Fragment-level interpretation seems intuitive; chemical features are encoded in fragments, rather than atoms themselves.

Weakness:

1. I have concerns on evaluation setups. Previous molecular property prediction works mainly utilize MoleculeNet, PCQM4Mv2 QM9 benchmarks. I think that evaluation on those benchmarks is essential to validate the effectiveness of FragFormer.

2. Analysis is weak. The authors claim the importance of non-overlapping fragmentation, but it seems that there is insufficient support on the claim. For example, analysis of fragments in different hierarchy is required.

Questions:

1. When the fragments are not in the vocabulary, how are they treated?

---

> ### Author Response · Authors · 2025-02-13
> **Review Response**
>
> We thank the reviewer for the encouraging and insightful comments. We have revised our manuscript according to these comments and suggestions. Please find our responses to specific questions and concerns below.
>
> **Question 1**: Previous molecular property prediction works mainly utilize MoleculeNet, PCQM4Mv2 QM9 benchmarks. I think that evaluation on those benchmarks is essential to validate the effectiveness of FragFormer.
>
> **Response**: We thank the reviewer for highlighting the importance of these benchmarks. We add the evaluation of Fragformer with DOVE-1 on the MoleculeNet benchmarks in the revised manuscript (Appendix F).
> We follow the experimental settings in KPGT[1]. We split the dataset into 80\% training, 10\% validation, and 10\% test with scaffold splitting.
> We report the mean performance over three runs with different random splits. The results are shown in the table below. All baseline results of atom-based models are taken from KPGT. FragFormer outperforms all fragment-based baselines and achieves the SOTA performance on 7 out of 11 datasets. We find that FragFormer excels at the target-related tasks, e.g., BBBP, BACE, Estrogen, while underperforms previous SOTA method on toxicity-related tasks, e.g., Tox21, ToxCast. We hypothesize that this could be due to differences in task characteristics, and we plan to investigate the underlying reasons for this phenomenon in future work. Although the performance of FragFormer is not the best on all datasets, it can additionally provide better fragment-level interpretation with FragCAM.
>
> | Method                  | BACE(CLS)  | BBBP(CLS)  | ClinTox(CLS)       | SIDER(CLS) | Estrogen(CLS)      | MetStab(CLS)      | Tox21(CLS) | ToxCast(CLS) | FreeSolv(REG)      | ESOL(REG)  | Lipo(REG)  |
> |-------------------------|-------|-------|---------------|-------|---------------|--------------|-------|---------|---------------|-------|-------|
> | Infomax                | 0.839 | 0.840 | 0.661         | 0.616 | 0.888         | 0.837        | 0.816 | 0.690   | 4.119         | 1.462 | 0.978 |
> | Edgepred               | 0.817 | 0.873 | 0.730         | 0.603 | 0.881         | 0.844        | 0.818 | 0.712   | 3.849         | 2.272 | 1.030 |
> | Contextpred            | 0.840 | 0.877 | 0.732         | 0.609 | 0.882         | 0.857        | 0.806 | 0.714   | 3.141         | 1.349 | 0.969 |
> | GraphLoG               | 0.830 | 0.846 | 0.667         | 0.615 | 0.871         | 0.850        | 0.796 | 0.677   | 4.174         | 2.335 | 1.104 |
> | GraphCL                | 0.825 | 0.887 | 0.691         | 0.587 | 0.875         | 0.821        | 0.805 | 0.696   | 4.014         | 1.835 | 0.945 |
> | JOAO                   | 0.826 | 0.879 | 0.741         | 0.640 | 0.861         | 0.837        | 0.823 | 0.711   | 3.466         | 1.771 | 0.933 |
> | GROVER                 | 0.840 | 0.887 | 0.874         | 0.638 | 0.892         | 0.876        | 0.838 | 0.696   | 2.991         | 0.928 | 0.752 |
> | 3DInfomax              | 0.811 | 0.877 | 0.887         | 0.585 | 0.880         | 0.866        | 0.805 | 0.716   | 2.919         | 1.906 | 1.045 |
> | GraphMVP               | 0.818 | 0.860 | 0.719         | 0.584 | 0.865         | 0.820        | 0.799 | 0.689   | 2.532         | 1.937 | 0.990 |
> | GraphMAE               | 0.857 | 0.878 | 0.748         | 0.597 | 0.881         | 0.853        | 0.801 | 0.691   | 3.023         | 1.378 | 0.746 |
> | KPGT                   | 0.855 | 0.908 | **0.946**     | 0.649 | 0.905         | 0.889        | **0.848** | **0.746** | 2.121       | 0.803 | **0.600** |
> | **Fragment-based**      |       |       |               |       |               |              |       |         |               |       |       |
> | GraphFP†              | 0.839 | 0.869 | 0.832         | 0.598 | 0.879         | 0.857        | 0.813 | 0.707   | 3.228         | 1.805 | 0.803 |
> | FraGAT†               | 0.813 | 0.841 | 0.726         | 0.577 | 0.867         | 0.841        | 0.771 | 0.682   | 4.049         | 2.043 | 0.913 |
> | PharmHGT†             | 0.846 | 0.891 | 0.895         | 0.625 | 0.892         | 0.874        | 0.824 | 0.715   | 2.411         | 1.276 | 0.779 |
> | FragFormer†           | **0.868** | **0.912** | 0.919 | **0.656** | **0.918** | **0.894** | 0.840 | 0.732 | **1.990** | **0.801** | 0.642 |
>
> (CLS represents a classification task, measured by AUROC, where higher values are better. The others are regression tasks, measured by RMSE, where lower values are better.)
>
> (Our work aims at utilizing the fragment (functional group) prior to improve molecular property prediction. For quantum property prediction like QM9, PCQM4Mv2, we think it is more suitable to use the 3D structure of the molecule which is beyond the scope of our work.)
>
> [1] Han Li et al., A knowledge-guided pre-training framework for improving molecular representation learning, Nature Communications 2023

---

> ### Author Response · Authors · 2025-02-13
> **Review Response**
>
> **Question 2** It seems that there is insufficient support on the claim that the importance of non-overlapping fragmentation. Analysis of fragments in different hierarchy is required.
>
> **Response**: We thank the reviewer for this suggestion. We further analyze the fragments in different hierarchy. We first construct three hierarchy of fragmentation: All Atom, Junction Tree (JT)[2], and DOVE-1. "All Atom" takes each atom as a fragment. JT decomposes the molecule into rings and bonds. The number of fragments follows $\text{All Atom} > \text{JT} > \text{DOVE}$.
> We then evaluate the performance of FragFormer with different fragmentation methods. The results are shown in table below. The overall performance of FragFormer with DOVE is better than JT, which is better than All Atom.
> Combined with the ablation study that DOVE-1 and DOVE-2 performs better than DOVE-0, we conclude that the fragmentation method DOVE and the non-overlapping feature is important for the performance of FragFormer. We also include these analysis in the revised manuscript (Appendix E) to strengthen the support.
>
> | Method / Dataset            |  AMES(CLS)  |  BBB(CLS)   |  CYP2C9  |  CYP2D6  |  CYP3A4  |  HLMC  |  MLMC  |  RLMC  |  LogD  |  PPB   |  Sol   |
> |--------------------|--------|--------|----------|----------|----------|--------|--------|--------|--------|--------|--------|
> | FragFormer-Atom    | 0.867  | 0.916  | 17.849   | 16.133   | 16.663   | 0.606  | 0.755  | 0.718  | 0.800  | 0.192  | 1.045  |
> | FragFormer-JT      | 0.863  | 0.919  | 17.413   | 15.666   | 16.603   | 0.550  | 0.746  | 0.711  | 0.709  | 0.185  | 0.938  |
> | FragFormer-DOVE    | **0.889** | **0.927** | **16.855** | **14.425** | **15.894** | **0.514** | **0.702** | **0.596** | **0.667** | **0.157** | **0.895** |
>
> **Question 3**: When the fragments are not in the vocabulary, how are they treated?
>
> **Response**: We thank the reviewer for pointing out this detailed but important question. If the fragments of interest are not in the vocabulary, they will be broken down into smaller fragments that exist within the vocabulary.
> Our method DOVE is based on iterative line graph and principal subgraph mining[1].
> It first learns a fragment library from the molecule database, and then decomposes the molecule into fragments based on the learned library. *So the produced fragments are always in the vocabulary in the standard process*. If expert knowledge suggests that certain fragments are particularly important, we can incorporate them into the fragment library or prioritize their merging during the decomposition stage.
>
> [1] Xiangzhe Kong et al., Molecule Generation by Principal Subgraph Mining and Assembling, NeurIPS 2022
> [2] Wengong Jin et al., Junction Tree Variational Autoencoder for Molecular Graph Generation, ICML 2018

---

> > ### Comment · Reviewer_BNJd · 2025-02-24
> >
> > Thank you for the responses. The responses have addressed my concerns. I do not have further questions at this moment.

---

### Review · Reviewer_8aDe · 2025-02-04

**Summary Of Contributions:**

- This paper introduces a novel method for representing molecular graphs using a k-degree overlapping fragmentation method (DOVE). This new representation aims to address a fundamental shortcoming of previous fragment based representation methods which cannot model an atom as a member of more than one functional fragment.

- To further enhance fragment based representation learning, this paper describes nested masked fragment prediction (MFP) which aims to capture hierarchical relationships since fragments can be nested. The masked prediction task involves predicting all masked substructures, not just one of them, as a self-supervised representation learning objective.

- A complementary model interpretation method, fragCAM is also introduced to help attribute predictions to functional fragments instead of atoms - this can enable a better analysis of which molecular substructures drive property predictions.

**Audience:**

Yes

**Claims And Evidence:**

Yes

**Requested Changes:**

The weaknesses listed above need to addressed to strengthen the work.

Critical changes: Discussion of the hyperparameter k in DOVE with experiments for larger values to establish a clearer trend and highlight the tradeoff between compute efficiency vs better performance so that practitioners can make an informed choice. Fidelity results for FragCAM are also required for tables 3 and 4.

Other changes: It is strongly recommended to include ablative experiments which study the improvements caused by DOVE/MFP in comparison to baseline method without DOVE and MFP while controlling for all other factors (such as fragment vocabulary).

Minor issues: The cited paper by Shrikumar et al. introduces DeepLIFT, not sure why it is referred to as GradInput in the text. Authors are requested to double check their references for accuracy.

**Strengths And Weaknesses:**

**Strengths**: Overall the paper is well written with detailed descriptions of their method and comprehensive experiments.

- DOVE overcomes the limitation of other methods being unable to model overlapping fragments using an iterative line graph method. This approach is strongly motivated by the domain and has theoretical justification, preserving the connectivity as well for these overlapping fragments.

- MFP is a well thought out method for self-supervised learning of hierarchical representations where fragments can be nested or overlapping and overcomes the simplified formulation of exclusive fragment membership for atoms in the molecular graph. This method can therefore capture structural information at varied scales as reflected by the diversity of fragment sizes.

- FragCAM is a helpful addition to enhance the attribution of molecular properties to functional fragments instead of atoms. Though its a simple formulation, experiments demonstrate its efficacy.

- Comprehensive experiments are conducted to compare FragFormer with existing molecular graph representation methods and demonstrate its effectiveness. Ablations are also conducted to isolate the improvements resulting from the modifications introduced by this method.

**Weaknesses**: There are a few points which could use some clarification.

- Choice of the hyperparameter **k** in DOVE: While experiments are conducted with k=0,1,2 it is unclear how to choose the best value of k for a given problem. The results in table 1 broadly show that k=1,2 works better than k=0 but there is variation across datasets and sometimes k=1 is better than k=2. While larger values of k may be more expensive computationally, I think it merits additional investigation to establish clear trends that can be used by practitioners to inform their choice of k.

- Fragment vocabulary construction's relationship with k - If a new vocabulary is being built for each k, it can be hard to isolate the improvements in the method to the vocabulary size vs the potentially more expressive graph structure. Ablations along this direction would be appreciated and be helpful to practitioners similar to the previous point.

- Knowledge fusion layer implementation: The choice of using a GRU to aggregate multiple knowledge features is confusing. GRU is sensitive to the ordering of inputs, so a shuffled ordering of the same features would lead to a different result. You may want to replace this layer with one which is permutation invariant like a self-attention layer without any positional encodings.

- FragCAM method: While this method is shown to be effective, CAM is an older interpretation method and there were better alternatives developed including DeepLIFT and Integrated Gradients which overcome some of its limitations. Both these are cited in the paper and are foundational to other graph based interpretation methods, so a justification for CAM (like efficiency / similar performance) would be nice.

- FragCAM evaluation: Table 3 and 4 only report AUROC, but it is critical to include the Fidelity metric (as used in SubgraphX) to accurately measure if the explanations are actually important to the model's predictions.

- Ablations: Given that FragFormer introduces multiple methodological improvements, the ablation studies only highlight the importance of each component as part of this larger system. While these experiments are insightful, it would be more helpful to take a baseline graph representation method and then introduce individual critical components like DOVE / MFP to demonstrate their utility over the traditional representation methods while controlling for vocabulary and pretraining method (if that is not the target of the ablation).

---

> ### Author Response · Authors · 2025-02-13
> **Review Response**
>
> We thank the reviewer for the encouraging and insightful comments. We have revised our manuscript according to these comments and suggestions. Please find our responses to specific questions and concerns below.
>
> **Question 1**: Discussion of the hyperparameter k in DOVE with experiments for larger values to establish a clearer trend and highlight the tradeoff between compute efficiency vs better performance so that practitioners can make an informed choice.
>
> **Response**: We thank the reviewer for suggesting a deeper exploration of the hyperparameter k.
> We further conduct experiments with $k=3$ on pharmabench and observe a performance drop compared to $k=2$. We hypothesize this decline may be due to excessive information sharing in the over-lapping regions, which make the pre-training task easier and less useful.
> The main computational cost difference lies in the tokenization stage for the pre-training datasets. In the table below, we provide the average time required to tokenize 1,000 samples from the ChEMBL29 dataset on a single CPU. As k increases, the computational cost for tokenization rises significantly.
>
>
> |              k              |  0  |  1  |  2  |   3  |
> |---------------------------|---|---|---|----|
> | time per 1k molecules (min) | 1.5 | 2.3 | 4.7 | 15.5 |
>
>
> Taking both the performance and computational efficiency into consideration, we recommend setting $k=1$ or $k=2$ in practice. We include the results of $k=3$ in the revised manuscript (table 1 and section 4.3 Results and Discussions).
>
> **Question 2**: Fragment vocabulary construction's relationship with k - If a new vocabulary is being built for each k, it can be hard to isolate the improvements in the method to the vocabulary size vs the potentially more expressive graph structure.
>
> **Response**: We thank the reviewer for raising this insightful point. Vocabulary construction inherently depends on k, as it is performed on $L^{k}(G)$. However, the vocabulary size is fixed as 500 across different k values in table 1 of the manuscript.
> We also conduct an ablation experiment to isolate the difference of vocabulary. We first build the vocabulary for $k=0,1,2,3$ with vocab_size=$500$ and merge the vocabularies to get the final vocabulary. Then, we pre-train the model for $k=0,1,2,3$ with the final vocabulary separately and evaluate the performance on Pharmabench.
> The results are shown in table below.
> The optimal results for each dataset occur at $k=1$ or $k=2$, which is consistent with
> the findings in table 1 of the manuscript.
> Interestingly, under the same vocabulary,
> $k=2$ yields superior performance more frequently than $k=1$.
> The trend is reversed in table 1 of the manuscript.
> We include this ablation study in the revised manuscript (Appendix E).
>
> | Method/Dataset |   AMES(CLS)   |   BBB(CLS)   |    CYP2C9   |  CYP2D6   |   CYP3A4   |   HLMC   |   MLMC   |   RLMC   |   LogD   |   PPB   |   Sol   |
> |----------------|----------|---------|-------------|-----------|------------|----------|----------|----------|----------|---------|---------|
> | FragFormer-0   |  0.858   |  0.918  |   17.358    |  17.241   |   16.859   |  0.625   |  0.854   |  0.603   |  0.780   |  0.193  |  0.952  |
> | FragFormer-1   |  0.865   | **0.922** |   17.316    |  14.833   |   15.773   | **0.517** |  0.734   |  0.565   |  0.695   | **0.175** |  0.910  |
> | FragFormer-2   | **0.870** |  0.920  | **16.732**  | **14.538** | **15.302** |  0.519   | **0.708** | **0.548** | **0.687** |  0.177  | **0.905** |
> | FragFormer-3   |  0.860   |  0.917  |   17.809    |  16.986   |   16.910   |  0.544   |  0.754   |  0.577   |  0.725   |  0.199  |  0.959  |
>
>
> (CLS represents a classification task, measured by AUROC, where higher values are better. The others are regression tasks, measured by RMSE, where lower values are better.)

---

> ### Author Response · Authors · 2025-02-13
> **Review Response**
>
> **Question 3**: Knowledge fusion layer implementation: The choice of using a GRU to aggregate multiple knowledge features is confusing. GRU is sensitive to the ordering of inputs, so a shuffled ordering of the same features would lead to a different result. You may want to replace this layer with one which is permutation invariant like a self-attention layer without any positional encodings.
>
> **Response**: We appreciate the reviewer for this insightful suggestion. Our design follows the message fusion operation in AttentiveFP[1] (equation 5 in [1]), which applies a GRU to agglomerates the messages.
> We fix the order of knowledge features in pre-training and fine-tuning, so the results are consistent. We also explore the self-attention layer to aggregate the knowledge features and find that it achieves similar performance as GRU (see table below, we adopt DOVE-1 in fragmentation). We will explore better knowledge aggregation methods in the future work.
>
> |                 |  AMES(CLS) |  BBB(CLS)  | CYP2C9 | CYP2D6 | CYP3A4 |  HLMC |  MLML |  RLMC |  LogD |  PPB  |  Sol  |
> |-----------------|:-----:|:-----:|:------:|:------:|:------:|:-----:|:-----:|:-----:|:-----:|:-----:|:-----:|
> |       GRU       | 0.889 | **0.928** | 16.855 | **14.425** | 15.894 | 0.514 | 0.702 | 0.596 | **0.667** | **0.157** | **0.895** |
> | Self-Attention | **0.890** | **0.928** | **16.747** | 14.579 | **15.784** | **0.511** | **0.696** | **0.594** | 0.677 | 0.167 | 0.901 |
>
>
> (We apply the self-attention layer to aggregate the knowledge features as follows. We first project the knowledge features to the same dimension as the fragment features. Assume the fragment vectors are $H \in \mathbb{R}^{N \times d}$, knowledge vectors are $G \in \mathbb{R}^{N \times d}$. We treat $F$ as query input, and $G$ as key and value input. Then, the self-attention mechanism is applied to query, key and value. The self-attention output is added with the fragment vectors $H$ to get the final representation.)
>
> **Question 4** While FragCAM is shown to be effective, CAM is an older interpretation method and there were better alternatives developed including DeepLIFT and Integrated Gradients which overcome some of its limitations. A justification for CAM (like efficiency / similar performance) would be nice.
>
> **Response**: We thank the reviewer for suggesting alternative interpretation methods. We adopt DeepLIFT and Integrated Gradients (IG) on the initial fragment vectors, which is the output of subgraph encoder, and denote these methods as FragDeepLIFT and FragIG. We set steps=50 for FragIG and use the default setting for FragDeepLIFT in Captum[2]. The results are shown in tables below. The fidelity and sparsity definition follows SubgraphX[3]. Higher fidelity under similar or lower sparsity are preferred in the evaluation. FragCAM outperforms FragDeepLIFT and FragIG on DrugXAI in terms of AUROC and fidelity, which demonstrates the effectiveness of FragCAM in explaining the prediction results with fragments.
>
> | AUROC    | FragCAM          | FragIG     | FragDeepLift |
> |--------------------|------------------|------------|--------------|
> | 3MR                | **0.981**        | 0.971      | 0.934        |
> | Benzene            | **0.935**        | 0.886      | 0.875        |
> | Mutagenicity       | **0.793**        | 0.711      | 0.704        |
>
> | Fidelity (sparsity)     | FragCAM        | FragIG       | FragDeepLift  |
> |--------------------|----------------|--------------|---------------|
> | 3MR                | **0.87(0.85)** | 0.87(0.78)   | 0.13(0.75)    |
> | Benzene            | **0.50(0.67)** | 0.46(0.64)   | 0.45(0.64)    |
> | Mutagenicity       | **0.35(0.72)** | 0.36(0.67)   | 0.19(0.66)    |
>
> We include this justification in the revised manuscript (Appendix C).
>
> [1] Zhaoping Xiong et al., Pushing the Boundaries of Molecular Representation for Drug Discovery with the Graph Attention Mechanism, Journal of Medicinal Chemistry 2020
> [2] Narine Kokhlikyan et al., Captum: A unified and generic model interpretability library for PyTorch, ArXiv 2020
> [3] Hao Yuan et al., On Explainability of Graph Neural Networks via Subgraph Explorations, ICML 2021

---

> ### Author Response · Authors · 2025-02-13
> **Review Response**
>
> **Question 5**  Fidelity results for FragCAM are also required for FragCAM and SubgraphX.
> **Response**: We thank the reviewer for pointing out the need for fidelity results, which helped us take a more comprehensive comparison between FragCAM and SubgraphX. We add the fidelity results for FragCAM and SubgraphX in the revised manuscript(table 4, related definition and discussion).
> FragCAM achieves the highest fidelity under similar or lower sparsity compared to SubgraphX.
>
> | $N_{min}$       | SubgraphX (3) | SubgraphX (7) | SubgraphX (10) | SubgraphX (15) | FragCAM         |
> |------------------|--------------|---------------|----------------|----------------|-----------------|
> | **AUROC**       | 0.655        | 0.683         | 0.663          | 0.592          | **0.793**       |
> | **Fidelity(sparsity)** | 0.24(0.81)   | 0.29(0.58)    | 0.30(0.41)     | 0.28(0.21)     | **0.35(0.72)**  |
> | **Time per sample (s)** | 52           | 28            | 17             | 11             | **0.024**       |
>
>
>
> **Question 6**: It would be more helpful to take a baseline graph representation method and then introduce individual critical components like DOVE / MFP to demonstrate their utility over the traditional representation methods while controlling for vocabulary and pretraining method.
>
> **Response**: We thank the reviewer for this constructive suggestion to further demonstrate the utility of key components. We begin with the baseline model, which employs mask atom prediction without incorporating knowledge fusion ('FragFormer-Atom w/o K'). Next, we introduce DOVE fragmentation with $k=1$ ('FragFormer w/o K & Nested MFP'). Subsequently, we incorporate the nested MFP pre-training criterion ('FragFormer w/o K'). Finally, we enhance the model by adding the knowledge fusion layer ('FragFormer'). The results are shown in table below.
> The results show that each component contributes to the performance improvement. We include this ablation study in the revised manuscript (Appendix E) to further demonstrate the utility of key components.
>
> |            Method/Dataset                         | AMES(CLS)                  | BBB(CLS)       | CYP2C9   | CYP2D6   | CYP3A4  | HLMC   | MLMC   | RLMC   | LogD   | PPB    | Sol    |
> |-------------------------------------|---------------------------|---------------|-------------|-------------|------------|------------|------------|------------|------------|------------|------------|
> | FragFormer-Atom w.o K               | 0.837                     | 0.913         | 18.264      | 16.691      | 17.190     | 0.620      | 0.996      | 0.937      | 0.894      | 0.236      | 1.387      |
> | FragFormer w.o (K&Nested MFP)       | 0.856                     | 0.921         | 17.919      | 15.659      | 16.894     | 0.563      | 0.877      | 0.774      | 0.740      | 0.211      | 1.134      |
> | FragFormer w.o K                    | 0.882                     | 0.921         | 17.851      | 14.815      | 16.055     | **0.512**      | 0.799      | 0.680      | 0.706      | 0.166      | 0.981      |
> | FragFormer                          | **0.889**                     | **0.928**         | **16.855**      | **14.425**      | **15.894**     | 0.514      | **0.702**      | **0.596**      | **0.667**      | **0.157**      | **0.895**      |
>
>
> **Question 7**: The cited paper by Shrikumar et al. introduces DeepLIFT, not sure why it is referred to as GradInput in the text. Authors are requested to double check their references for accuracy.
>
> **Response**: We thank the reviewer for the thoroughness. We follow the citation convention for GradInput in previous work, e.g., [2][3].
> We guess the reason is that [1] also describes GradInput in the article.
> We have also checked the correctness of other references.
>
> [1] Avanti Shrikumar et al., Learning Important Features Through Propagating Activation Differences, ICML 2017
> [2] Marco Ancona et al., Towards better understanding of gradient-based attribution methods for Deep Neural Networks, ICLR 2018
> [3] Huiqi Deng et al., Unifying Fourteen Post-Hoc Attribution Methods With Taylor Interactions, TPAMI 2024

---

> > ### Comment · Reviewer_8aDe · 2025-02-21
> >
> > I appreciate the comprehensive follow-up experiments and clarifications - all my questions have been addressed.

---

### Decision · Action_Editor_6eU7 · 2025-03-29

**Recommendation:** Accept as is

**Comment:**

The paper proposes FragFormer, a self-supervised framework for molecular representation learning based on molecular fragments and masked fragment prediction. All the reviewers find the paper well-written, technically sound, and supported by comprehensive experiments. As the concerns of the reviewers have been also well addressed, the AE recommends accepting the paper.

**Audience:**

Yes.

**Claims And Evidence:**

Yes.

---

> ### Author Response · Authors · 2025-04-18
>
> We sincerely appreciate the insightful feedback offered by all the reviewers and the action editor.